# Coffee Berry Borer (*Hypothenemus hampei*), a Global Pest of Coffee: Perspectives from Historical and Recent Invasions, and Future Priorities

**DOI:** 10.3390/insects11120882

**Published:** 2020-12-12

**Authors:** Melissa A. Johnson, Claudia Patricia Ruiz-Diaz, Nicholas C. Manoukis, Jose Carlos Verle Rodrigues

**Affiliations:** 1Daniel K. Inouye US Pacific Basin Agricultural Research Center, United States Department of Agriculture—Agricultural Research Service, Hilo, HI 96720, USA; nicholas.manoukis@usda.gov; 2Oak Ridge Institute for Science and Education, Oak Ridge Associated Universities, Oak Ridge, TN 37830, USA; 3Department of Agroenvironmental Sciences, Center for Excellence in Quarantine & Invasive Species, Agricultural Experimental Station—Rio Piedras, University of Puerto Rico—Mayaguez, 1193 Calle Guayacan, San Juan, PR 00926-1118, USA; claudia.ruiz2@upr.edu (C.P.R.-D.); jose.rodrigues@upr.edu (J.C.V.R.)

**Keywords:** *Coffea arabica*, biocontrol, biosecurity, cultural control, integrated pest management, invasion biology

## Abstract

**Simple Summary:**

Coffee berry borer (CBB) is the most serious insect pest of coffee worldwide, causing more than US$500M in damages annually. Reduction in the yield and quality of coffee results from the adult female CBB boring into the coffee fruit and building galleries for reproduction, followed by larval feeding on the bean itself. This review examines the invasion biology of CBB in various coffee-growing regions throughout the world, comparing and contrasting patterns in historically invaded countries with those that were more recently invaded. The situation in Hawaii is highlighted as a case study for the development and implementation of a successful integrated pest management (IPM) program following 10 years of research and outreach.

**Abstract:**

Coffee berry borer (*Hypothenemus hampei* (Ferrari), CBB) has invaded nearly every coffee-producing country in the world, and it is commonly recognized as the most damaging insect pest of coffee. While research has been conducted on this pest in individual coffee-growing regions, new insights may be gained by comparing and contrasting patterns of invasion and response across its global distribution. In this review, we explore the existing literature and focus on common themes in the invasion biology of CBB by examining (1) how it was introduced into each particular region and the response to its invasion, (2) flight activity and infestation patterns, (3) economic impacts, and (4) management strategies. We highlight research conducted over the last ten years in Hawaii as a case study for the development and implementation of an effective integrated pest management (IPM) program for CBB, and also discuss biosecurity issues contributing to incursion and establishment. Potential areas for future research in each of the five major components of CBB IPM (monitoring and sampling, cultural, biological, chemical, and physical controls) are also presented. Finally, we emphasize that outreach efforts are crucial to the successful implementation of CBB IPM programs. Future research programs should strive to include coffee growers as much as possible to ensure that management options are feasible and cost-effective.

## 1. Introduction

Coffee berry borer, *Hypothenemus hampei* (Ferrari) (Coleoptera: Curculionidae: Scolytinae) or CBB, is the most damaging insect pest of coffee worldwide, affecting both the yield and quality of coffee products [1,2] and causing more than US$500 million in damage annually [3]. Damage to the marketable coffee product occurs when the female beetle (Figure 1A) bores a hole into the coffee fruit (often referred to as a berry) and eventually into the seed (or “bean”) (Figure 1B), where she builds galleries for reproduction (Figure 1C), followed by larval feeding on the endosperm [4,5] (Figure 1D). In addition to being a serious pest of coffee, CBB is cosmopolitan today: over about 100 years, CBB has spread from the humid evergreen forests of Africa to nearly all coffee-producing countries in the world (the exceptions being Australia [6], China, and Nepal [5,7,8]) (Figure 2).

As with other insects of economic importance, research is the key to improved management of CBB. However, there has been little peer-reviewed research published on CBB to date relative to other globally important crop pests. From a cumulative literature search that included “*Hypothenemus hampei*” in the title and/or abstract we found 440 publications in the Dimensions database (https://www.dimensions.ai; Digital Science and Research Solutions Inc. 2020) (Figure 3A). Dimensions is a large database of more than 100 million publications, ranging from articles published in scholarly journals, books and book chapters, to preprints and conference proceedings and grants, patents, clinical trials, datasets and policy documents, all indexed and linked by a leading Artificial Intelligence (“AI”) algorithm. A similar search in the more inclusive Google Scholar database recovered 2680 publications (Figure 3A). The synonyms of *H. hampei* (*Cryphalus hampei, Stephanoderes hampei, Stephanoderes coffeae, Xyleborus coffeicola, Xyleborus coffeivorus, Hypothenemus coffeae*) were not broadly used, and given that these represented a small fraction of the total publications they were not included in our combined data. In comparison, a literature search in Dimensions (*D*) and Google Scholar (*GS*) on the major crop pests *Helicoverpa armigera* (cotton bollworm), *Bemisia tabaci* (tobacco whitefly), and *Ceratitis capitata* (Mediterranean fruit fly) resulted in *D* = 3726, *GS* = 9830; *D* = 4311, *GS* = 9750; and *D* = 2445, *GS* = 5330 publications, respectively. This discrepancy may be partly due to the fact that CBB has a single host genus unlike many other major crop pests. However, this is offset by the large role that coffee production plays in many (especially developing) economies in the tropics. Finally, our Dimensions literature search revealed that the majority of publications focusing on CBB over the last 10 years describe the biology and ecology of this pest, and potential biological and chemical controls (Figure 3B).

Following a brief review of CBB taxonomy, origin, biology, and distribution, we examine the status of this pest from a global perspective to assess similarities and differences among regions invaded in the early stages of its spread and those that were recently invaded. Specifically, we detail the invasions, detection, and responses in three key coffee-growing countries that were invaded early in the spread of CBB (Indonesia, Brazil, and Ethiopia) and contrast them with perspectives on what is being learned from three of the most recently invaded regions: Puerto Rico (Caribbean Sea/Atlantic Ocean) in 2007, Hawaii (Pacific Ocean) in 2010, and Papua New Guinea (Indian Ocean) in 2017 (Figure 2). Lastly, we focus on Hawaii as a case study for the development and implementation of an effective integrated pest management program for CBB following 10 years of research and outreach to growers.

## 2. Taxonomy and Identification

The genus *Hypothenemus* is one of the largest genera within the Scolytidae family (“bark beetles”) with 181 described species [9,10,11,12]. The majority of *Hypothenemus* species are poorly known, being very small (<2 mm long) wood-boring beetles that occur in tropical and subtropical areas [13]. The coffee berry borer was first described as *Cryphalus hampei* by Ferrari in 1867 [14] from specimens obtained in traded green coffee beans that were imported to France from an unknown location [15]. Following Eichhoff’s [16] description of the genus, the species was later moved to *Stephanoderes*, a genus treated as a synonym of *Hypothenemus* by Swaine [17]. The genus *Stephanoderes* was eventually moved to *Hypothenemus* by Browne [18], with *Hypothenemus hampei* being the currently accepted name for the species.

*Hypothenemus hampei* is distinguished from other *Hypothenemus* species by the following morphological characters: a narrowly rounded pronotum with mixed setae and some slightly flattened (Figure 4A,B); a broadly rounded elytral declivity without a distinct transition from the elytral disc (Figure 4B); prominent interstrial bristles in uniseriate rows, and the bristles being long, narrow, and slightly flattened (Figure 4B); a broad, indistinctive frontal grove or no groove at all on the frons (Figure 4C); usually four marginal asperities [5]. Hopkins [19] noted that one of the defining characters for taxa within the genus *Hypothenemus* is a four-segmented antennal funicle, while five segments is characteristic of the genus *Stephanoderes*. However, this character is unreliable as some species can have a range of 3–5 segments [5], including *H. hampei* (see Figure 4C). This suggests that a detailed taxonomic review of the genus should be conducted, taking into account relevant information on morphology and phylogeny.

## 3. Origin

CBB was first reported in coffee plantations in Liberia in 1897 [19], followed by the Democratic Republic of Congo in 1901 [1] and Zaire in 1903 [21]. The geographic origin and initial host of CBB within Africa remain unclear. Some authors have speculated that it emerged in Ethiopia in conjunction with *Coffea arabica* L. (commercially known as arabica; Rubiaceae), thriving at altitudes from 1200 to 2000 m [22]. However, Davidson [23] concluded in 1967 that the pest was not present in Ethiopian coffee farms, and it was not until 1998 that CBB was reported to be a problem in the country [24]. Prior to 1984, low annual minimum temperatures in the area around Jimma, Ethiopia (1780 m) would have prevented CBB from completing a single generation per year [25], suggesting that the original host was likely the low-altitude species *Coffea canephora* Pierre ex Frohner (commercially known as robusta), which occurs from 250–1100 m in West and Central Africa [4,25,26,27]. Under this scenario, CBB could have spread from infested robusta coffee berries originating in West Africa to arabica coffee in Ethiopia or Saudi Arabia, where it was first imported before the 15th century [21].

## 4. Biology and Life History

CBB is distinguished from all 850 other insect species that can feed on parts of the coffee plant in that it is the only one able to feed and complete its life cycle in the coffee seed, thought to be possible by its association with *Pseudomonas* bacteria in its alimentary canal that acts to detoxify caffeine [28]. Adult female CBB colonize coffee berries when the dry weight content of the coffee seed is 20% and the endosperm is in the state of development known as semi-consistent [29], approximately 120–150 days after flowering [2,30]. A small entrance hole (0.6–0.8 mm wide [31]) is typically bored near or through the floral disc (Figure 1B), which provides an ideal rough surface for the beetle to grasp while boring [32,33]. Usually, there is only a single hole per berry unless the infestation is high and the availability of berries is scarce, in which case multiple females may bore entrance holes into the berry [34,35,36]. Laboratory studies have estimated that the time required to enter the berry is ≈2–8 h [34,37,38]. Four stages describe the position of the colonizing female within the berry: (A) the female has initiated penetration of the exocarp, (B) the female has penetrated the endocarp, and only part of the abdomen is visible, (C) the female is no longer visible and has bored into the endosperm, and (D) the female has constructed galleries and is reproducing within the seed [30].

Females can lay >100 eggs [25,39] over an oviposition period of 11–40 days [25,40]. The average time to complete the life cycle from egg to adult depends largely on temperature and berry moisture, with faster development times as temperatures increase and berry moisture decreases [41,42,43,44]. The sex ratio is skewed towards females, but estimates vary widely among published studies (5:1 to 494:1; reviewed in [5]). Sex determination in CBB is described as functional haplodiploidy; both males and females are diploid, but males are functionally haploid due to the paternal set of chromosomes condensing into a mass of chromatin and are therefore unincorporated or nonfunctional in somatic cells [45]. Males are smaller than females (0.99–1.3 mm long vs. 1.6–1.9 mm long) and have smaller eyes and wings [4,39]. There is a high degree of inbreeding, with almost all mating occurring between siblings. Male and female siblings mate within the berry; the males die, and the fertilized females leave their natal berry to find a new berry in which to deposit their eggs [41]. The number of CBB generations per year has been estimated to vary between 2 and 13, depending on the climatic conditions at a given location [5,25,43,44,46,47].

## 5. Worldwide Pest Status

Molecular studies have been used to track the spread of CBB from Africa to other coffee-producing countries [48,49], and to determine which regions are likely source areas for new introductions [50,51,52,53]. Benavides et al. [50,52] used AFLP (amplified fragment length polymorphism) fingerprinting to analyze genetic variability and biogeographic patterns based on samples from 17 countries. These authors concluded that there were three separate introductions of CBB into the Americas and that West Africa was likely the origin of introductions into America and Asia. Gauthier [54] sampled CBB populations across Africa, Asia, Central America, and South America, and used the mitochondrial COI gene to conduct a Bayesian clustering analysis that revealed five distinct populations: (1) Ethiopia, (2) Kenya and Uganda, (3) Brazil, (4) Central America, Colombia, and the Dominican Republic, and (5) Indonesia, New Caledonia, India, West Africa, and Jamaica. As of 2017, CBB has successfully invaded all coffee-producing countries in the world, except for China, Nepal, and Australia (Figure 2). CBB likely entered most new regions either through infested green coffee beans which were not adequately treated before importation [54,55] or through plantation workers that unknowingly brought the pest in clothing or equipment used on infested farms [53].

## 6. Insights from Historical Invasions: Indonesia, Brazil, Ethiopia

### 6.1. Social and Economic Constraints Are Preventing Implementation of IPM: Indonesia as a Case Study

Indonesia is among the world’s top coffee producers and exporters, ranking fourth in both categories in 2016–2017 [56]. Coffee has been cultivated in Indonesia since 1699 (more than 300 years), primarily by smallholding farmers (96%) [57]. Coffee production in the country is dominated by robusta (80% of all coffee produced), which is grown primarily in the southern Sumatran provinces of Lampung, South Sumatra, and Bengkulu; and Java and Flores [58]. Arabica (19%) and liberica (1%) are grown to a lesser extent, primarily in North Sumatra, Aceh, and Java [59]. Indonesian coffee is mostly grown under permanent shade trees, and inter-cropping is common to obtain additional income [57]. Coffee trees are typically managed using the single stem pruning system, and inorganic fertilizers and/or organic fertilizers are applied at low levels [58], while pesticides are very rarely used [57]. Indonesian coffee is produced with fewer agri-chemical inputs than many other major coffee-producing countries (e.g., Brazil, Colombia, and Vietnam), possibly due to the high prices of these imported products, set by the government [58].

The first report of CBB in Indonesia was in 1908 from *Coffea liberica* growing in West Java [60]. By 1919, all of the West Java region was infested and the pest continued its spread to East Java, Sumatra, Bali, Flores, Kalimantan, Sulawesi, and Papua [61,62]. Wiryadiputra et al. [63] estimated that the average CBB infestation across the country was 20% (range 10–50%). National yield losses are estimated to be ≈10% [63], or a minimum of 62,500 tons of green coffee (US$100 million/year) [57]. Surveys in the Gayo Highlands of Aceh in Sumatra showed that low-elevation farms (950–1250 m) had considerably higher infestation than mid-elevation (1250–1400 m) and high-elevation (>1400 m) farms [57]. Field observations in 2005 suggested that infestation was highest from the beginning of the harvest (July) through to the post-harvest season (January) [57]. Peaks in flight activity lagged slightly behind but were largely coincident with infestation trends [57].

The development and implementation of an effective IPM strategy for CBB in Indonesia is ongoing, with several significant components being explored: (1) sanitation of tree and ground raisins for areas with short growing seasons, and large estates, (2) pruning of both coffee and shade trees, (3) natural predators such as birds and ants, (4) parasitoids including the previously released *Prorops nasuta* and *Cephalonomia stephanoderis*, (5) sprays of the entomopathogenic fungus *Beauveria bassiana*, and (6) monitoring and control through mass trapping using alcohol-baited funnel traps [57,63,64].

Poor management of smallholder coffee farms due to insufficient training of farmers, plus economic constraints, are serious obstacles to coffee production and CBB control. Saragih [59] found that arabica (which draws a higher price on the world market) production and quality could be increased in Indonesia via land use optimization, the use of family labor, and the application of good agricultural practices (GAPs), such as the planting of shade trees, pruning, using organic fertilizer at recommended rates, land conservation, and control of CBB. While large estates have the resources to manage CBB using intensive sanitation practices, smallholder farms are often unable to afford these methods due to labor costs [57,64]. Pruning can rejuvenate coffee trees and increase production, but can have differing effects on CBB infestation and dispersal depending on the technique used. A recent study examining the effects of pruning unproductive branches and suckers on coffee trees in the Toba highlands of North Sumatra showed that yields increased significantly in pruned vs. unpruned trees. Importantly, CBB infestation was not increased by this pruning method, and remained similar in the two treatments [65]. Thus low-intensity pruning could be a way for smallholder farms to increase their production while keeping CBB infestation at manageable levels [65]. High-intensity pruning practices (e.g., stump pruning by block) that have been shown to significantly reduce CBB populations [66,67,68] could be employed by large estate growers that can afford to have reduced production for a year.

### 6.2. Chemical Controls Are Not Sustainable: Brazil as a Case Study

Coffee seeds were first brought to Brazil from French Guiana and planted in the state of Pará in 1727 [69]. Coffee plantations quickly expanded to Rio de Janeiro, São Paulo, and Minas Gerais [70], with these three states accounting for 20% of the world’s production by the 1820s [71]. In the 1840s Brazil became the world’s largest coffee producer, a distinction it has held for more than 150 years [72]. Coffee production continued to expand into the 20th Century, reaching southern Minas Gerais, Espirito Santo, Parana, and Rondonia [72]. The Brazilian economy was tightly linked to the coffee industry during this long expansion period, and the coffee market was thus highly regulated by the federal government. The state-run Brazilian Coffee Institute (IBC) was responsible for maintaining favorable international prices by regulating supply and demand [73]. Coffee beans produced in Brazil were stockpiled, with lower-grade beans supplied to roasters for sale in the domestic market, and higher-quality beans exported [72,74]. During the export-promoting period of the 1960s and 1970s, many credits and subsidies were provided to growers as an incentive to increase coffee cultivation, and state-funded research was conducted to improve yields and disease resistance [75].

Following the breakdown of the international coffee agreement (ICA) in 1989, the global market was flooded with coffee reserves, driving the price of coffee down [73]. Upon the dissolution of the IBC in 1990, the Brazilian coffee sector changed dramatically, as producers raced to modernize and intensify coffee production [72]. However, market liberalization, deregulation, and globalization ultimately favored foreign interests, corporations, and large landholders with access to capital [76,77]. In contrast, small-scale landholders and producers experienced declining returns due to their dependence on largely unpredictable world coffee prices [78].

In Brazil, coffee is grown primarily by small-scale farmers in impoverished areas that are rural and isolated [73]. The steep topography, nutrient-poor soils, and limited access to markets have resulted in coffee being one of few crops grown on a large scale in these areas [79]. Brazilian coffee plantations are grown under full sun to maximize yields [73]; this system of full-sun cultivation has also been promoted as a means of limiting outbreaks of coffee leaf rust, which was purported to be more prevalent under shade [80]. Coffee trees are planted at high densities (3000–5000 trees/ha) on steep, sometimes vertical, bare slopes [73]. Soil erosion can reduce the average size [80] and productive lifespan [81] of coffee trees grown under this cultivation system by 50%, after which growers are forced to either abandon the land, convert it into cattle pasture, or illegally expand the size of their fields by annual small-scale burning of neighboring protected forest areas [79,82]. The combination of little to no government assistance, geographic isolation, poor soils, and steep terrain means that growers are seldom able to diversify their crop production, resulting in a continued cycle of high-density monoculture coffee plantings on degraded soils [73].

CBB was first introduced to Brazil in 1913 in Campinas, state of São Paulo, through green coffee beans imported from the Democratic Republic of Congo [83]. However, plantation workers did not initially realize that it was CBB, and its status as a severe pest was not recognized until 1924 when it had become widespread throughout the coffee-growing region [84,85]. In response to the invasion, the São Paulo state government formed a scientific commission to control the pest [86]. The campaign to control CBB included research, monitoring, and an educational film produced in 1925 that described CBB biology, identification, damage, and methods of control; the film was shown in cinemas 232 times and viewed by 104,634 people [86]. Recommendations for CBB control made by the commission included: (1) destroy all coffee trees in abandoned plantations, (2) remove all berries from the trees and ground after harvest, and burn or bury collected berries at least 12 inches deep, (3) disinfect harvested berries and collection sacks with carbon disulfide, and (4) examine all workers coming from other plantations for infested berries. Following the implementation of these strict control measures, some districts reported that infestation fell from 90–100% down to 1–5%. However, due to the high costs associated with fumigation and sanitation, many growers either went bankrupt or shifted from coffee cultivation to growing cotton or corn [87]. In 1929, the parasitoid *Prorops nasuta*, a natural enemy of CBB that parasitizes larvae and pupae, was imported to Brazil from Uganda and subsequently released as a biological control [88]. Once it was confirmed that the parasitoid could increase under field conditions, there was widespread demand and small colonies were distributed among growers [87]. However, the long-term establishment of the parasitoid in the field as a classical biological control was not consistent, rather multiple releases were required throughout the coffee season to maintain parasitoid populations [5,89]. Despite the enthusiasm for the project and efforts by growers to rear their parasitoids [87], these augmentative biological control efforts were suspended in Brazil with the development of chemical controls in the mid-20th century [90].

In 1947, Brazil initiated the use of synthetic insecticides against CBB [91]. The broad-spectrum contact and ingestion organochlorine insecticide endosulfan became the most effective and widely used chemical control against the pest in the 1960s, with just a single application reducing infestation by up to 88% and providing protection for up to 12 weeks [92]. Around the 1970s, coffee leaf miner (CLM), *Leucoptera coffeella* (Guérin-Méneville), was considered the primary coffee pest in some Brazilian regions because CBB was under chemical suppression by the widespread use of endosulfan. However, by 2013 endosulfan was banned in at least 70 countries (including Brazil) and continues to be phased out in others due to environmental and human health concerns, and growing evidence for CBB resistance to the insecticide [93,94,95,96]. The banning of this product and the pressure for human and environmental protection has renewed interest in sustainable alternatives for both pests.

While less toxic chemical alternatives to endosulfan exist (e.g., fenitrothion, fenthion, pirimiphos methyl-methyl), they are often not as effective at reducing CBB populations [97]. The use of systemic products such as thiamethoxan to control CLM, and its accumulation in the fruit tegument may present only a partial control of CBB [98]. A newer combination pesticide that is registered for use in Colombia and is a mixture of chlorantraniliprole and thiamethoxam, appears to be effective [99,100], but given that Colombia and Brazil are quite different in terms of microclimate conditions, these pesticides may not provide the same level of control in both countries. Furthermore, the environmental conditions that favor CBB outbreaks are not necessarily the same that will be preferred by CLM. Perhaps most importantly, the cost of these chemicals is too high for many farmers. Cheaper and more sustainable alternatives that have CBB repellant properties (e.g., neem oil, kaolin clay, garlic) are commercially available, but field tests have shown that these products are only effective for short periods due to rapid degradation [101,102].

The ban on endosulfan coupled with ideal environmental conditions for CBB reproduction in 2017 resulted in the worst CBB situation in recent memory, with bean damage estimated to range from 5–30% [103]. Growers with high levels of CBB infestation often have difficulty finding buyers [103]. Brazil’s Coffee Roasters Association (Brazilian acronym, ABIC) currently recommends that processors check samples of green coffee to ensure that bean damage does not exceed 5% [103]. Economic losses due to CBB in Brazil have been estimated at $215–358 million annually [104]. CBB population sizes in Brazil appear to be negatively correlated with rainfall intensity, with the highest population growth occurring during the dry season (May–August) [105,106]. CBB monitoring in robusta (known as conilon in Brazil) [107,108] and arabica [109] coffee plantations have revealed similar infestation patterns, with peak populations observed at the end of the harvest season (July–August). Studies elsewhere have also shown a positive correlation between trap catch and berry infestation [110,111], suggesting that traps may be a more cost-effective way to monitor CBB populations and determine an appropriate action threshold or economic injury level that would serve to inform the timing of pesticide applications [106]. Additionally, several studies have suggested that alcohol-baited traps may be an effective method of control if used at a high density (22–25 traps/ha) [64] and at the appropriate time in the season when CBB females are actively flying (post-harvest to early fruit growth) [112,113].

### 6.3. Climate Change Is Driving CBB’s Invasion Success: Ethiopia as a Case Study

Ethiopia is the main producer of coffee in the African continent, and the fifth largest exporter of arabica coffee globally [56]. Coffee is the backbone of the Ethiopian economy, accounting for 70% of the foreign exchange earnings and 10% of the government revenue while employing 25% of the domestic labor force [114]. Four coffee production systems are used in Ethiopia: coffee gardens tended by smallholder farmers near their residences (70%), semi-forest and forest coffee (25%), and modern plantations (5%) [115]. Smallholder producers account for 95% of production, while state-owned plantations and investor plantations account for 4.4% and 0.6% of production, respectively [116]. Arabica coffee originated in Ethiopia, and only grows in its wild form in Ethiopia, Uganda, and Kenya [117]. The top coffee-producing districts in Ethiopia are Oromia, South Nations, Nationalities, and Peoples Regional State [118]. The major constraints to coffee production in Ethiopia include diseases, insect pests, a lack of access to market information, lack of physical infrastructure, poor extension services, limited farm management, low soil fertility, and changes in climatic conditions [119].

CBB was initially described in Ethiopia from several reports of damaged berries in the southwest of the country in the 1970s and 1980s [120,121,122]. However, it was not until Abebe [24] reported the pest to be present in all but one site in the coffee-growing regions of the south and southwest that CBB was recognized as a significant problem. A study by Mendesil et al. [35] conducted at 14 localities in southwest Ethiopia reported that CBB was present at sites ranging from 1200–1700 m asl, and that small-scale farms generally had low infestation (<5%) relative to large-scale plantations and research sites (20–60%). The authors suggested that this result could be related to a longer history of CBB establishment in these areas, in addition to changes in cultural practices associated with newly planted varieties that may have created more favorable conditions for CBB reproduction and survival. The abundance of CBB was also reported to coincide with the availability of dried raisin berries left on trees and the ground after harvest, and the persistence of these dried berries appeared to contribute to the occurrence of CBB in coffee plantations year-round [35]. Asfaw et al. [123] examined infestation at three sites in southwestern Ethiopia that varied in altitude and production systems and reported significantly higher levels of damage at the low-elevation site relative to the mid-and high-elevation sites, and more significant damage in plantation systems compared to gardens.

The management of CBB in most African countries, including Ethiopia, has mainly involved cultural control methods that are inadequate alone, or too costly to perform consistently [124]. Chemical controls are used in some instances, but farmers often use insecticides only if advised by qualified extension workers; high costs and low availability of insecticides keep most smallholder farmers from using chemical controls [124]. Increased outreach and education on CBB awareness and management strategies, and further research on biological controls and coffee varietal resistance, are top priorities for CBB management in Africa [124].

Research by Jaramillo et al. [18] suggested that the rapid population growth and spread of CBB in Ethiopia in recent years are directly related to increasing temperatures. From 1979–2005, the Ethiopian highlands experienced a temperature increase of 0.39–0.65 °C [125]. Prior to 1984, temperatures in the Ethiopian highlands were cool enough to prevent CBB from completing a single generation per season/year, but it is now estimated that there are 0–2 generations per season/year [25]. Jaramillo et al. [25] predicted that just a 1–2 °C increase would shorten CBB development times sufficiently enough to allow additional generations per season and the expansion of the pest’s geographic range, while an increase above 2 °C would lead to shifts in the pest’s altitudinal and latitudinal range. The changes predicted by Jaramillo et al. [25] already seem to be occurring, with CBB reported at higher altitudes than previously found (>1800 m asl) in East Africa [126]. A follow-up study by Jaramillo et al. [8] predicted that the number of generations per season/year would increase along an altitudinal gradient in response to rising temperatures. Current estimates for East Africa are 1–4.5 generations per season/year, but by 2050 is predicted to increase to 5–10 generations per season/year for high-elevation farms (1400–1800 m asl) and 11–16 generations per season/year for low and mid-elevation farms (900–1300 m asl). Furthermore, Moat et al. [127] suggested that depending on the emissions scenario the bioclimatic area suitable for arabica cultivation could decline by ≈39–59% by the end of the century.

Planting shade trees in coffee plantations has been suggested as a potential strategy to mitigate the impacts of rising temperatures on coffee quality and production [8]. Shade trees provide several benefits to coffee plantations, including erosion control, windbreaks, increased nutrient cycling and organic matter, temperature buffers, and increased income for growers by planting fruit or timber trees [5]. However, arguments against using shade trees in coffee plantations include that they may lead to higher infestation levels of CBB and reduced coffee yields. Although several studies have found higher CBB infestation in shade vs. sun coffee [34,112,128,129], other studies have found significantly lesser CBB infestation [130,131] and population densities [112,132] in shade vs. sun-grown coffee, likely due to changes in abiotic factors (decreased temperatures and increased relative humidity), variance in agronomic practices, and increased natural predator density (e.g., parasitoids, birds, and ants). Higher yields [130], lower yields [133], and unaffected yields [134,135] have also been reported in shade vs. sun coffee. The variation in results across studies suggests that individual coffee-growing regions should further examine various species of shade trees and levels of shade to determine how best to use shade to mitigate CBB impacts under future climate scenarios.

### 6.4. Summary: What Have We Learned from Historical Invasions?

The issues related to coffee production and CBB management highlighted above are not specific to these early-invaded regions, they apply to all coffee-growing countries of the world. Several collective themes are summarized here. First, the vast majority of coffee is grown by smallholder farmers, often in impoverished areas. These farmers often have limited access to the education, tools, materials, and labor that is necessary to keep CBB infestations at manageable levels. This leads to a cycle of producing low-quality coffee that in turn receives low prices on the world market. Second, smallholder coffee farmers have historically relied on government assistance to grow their crop, and while this has helped farmers in the short term, the long-term result of government reliance is the inability and/or willingness of farmers to take the steps necessary to improve their management practices and ensure that their crop is sustainable into the future. Examples of this include the heavy reliance on chemical pesticides to control CBB in many regions of the world, and the shift from growing coffee in its natural state as an understory tree interspersed with other plants, to a predominantly sun-grown monoculture crop in order to intensify yields. Lastly, changes in how coffee is grown throughout the world are important because research from many coffee-growing countries now shows that climate change is having or will soon have significant impacts on the coffee industry. The area suitable for coffee cultivation (particularly for arabica coffee which prefers higher elevations) is predicted to decrease dramatically in the near future as global temperatures continue to rise. Rising temperatures are not only predicted to decrease suitable growing area for coffee, but will also decrease insect development times, leading to increased pest pressure.

## 7. New Perspectives from Recent Invasions: Puerto Rico, Hawaii, Papua New Guinea

### 7.1. Puerto Rico

Coffee was introduced to Puerto Rico (PR) in 1736 from Santo Domingo (Dominican Republic) [136], with the island’s coffee industry becoming established in the 1880s [137,138]. By the end of the 19th century, Puerto Rican coffee was recognized as some of the best in the world [138]. Coffee production has historically been the most important economic and cultural activity in PR. Coffee is currently the fifth most valuable agricultural crop in PR [139], and the largest contributor to the economy of the central mountain region of the island [140]. The majority of farms in Puerto Rico are smallholder operations <12 ha in size [141]. Coffee is planted on average across 60% of the farm land, with the remaining land being forested or inter-cropped with bananas, plantains, or citrus, most of which is not sold but used for personal consumption [141]. From 1982–2007 there was a 70% decrease in shade-grown coffee, a movement which was first encouraged by the government in the 1960s in order to increase yields [142]. Most growers receive some form of governmental assistance, including help with paying salaries to employees, access to machinery, the supply of fertilizer, and even assistance with herbicide application [141]. The average age of coffee farmers surveyed in 2012 was 58 years old, with most having been raised on coffee farms [141].

Socioeconomic factors such as population migration from coffee-growing areas, high production and labor costs, low availability of workers, recurrent destruction by tropical storms and hurricanes, and the arrival of pests and diseases (namely coffee leaf miner *Leucoptera coffeella* and coffee leaf rust *Hemileia vastratix*) have all contributed to the loss of substantial profits from coffee in recent decades [143,144,145,146]. Puerto Rico had 9805 coffee farms in 2002, but just five years later had plummeted by 40% to 5678 coffee farms remaining on the island [147]. This represented an economic decline of 10% throughout the PR coffee industry [147]. These declines were primarily the combined result of the fixed price of coffee sales imposed by the government, a scarcity of farm laborers, and increased production costs [146].

The presence of CBB was first reported in Puerto Rico in the mid 1940′s by Nolla [148] and Roque [149]. However, Wolcott [150] concluded in 1948 that the pest found in coffee fruits was in fact another Scolytid beetle, possibly *Hypothenemus seriatus*, which is one of several species known as “false coffee berry borer” that can enter the coffee berry but are not able to feed on the seed. In 1995, CBB was reported in the neighboring islands of the Dominican Republic [5]. At that time, initiatives such as quarantine mandates to prevent its introduction to Puerto Rico were developed, and educational extension initiatives to alert growers to the imminent threat to the industry. Vega et al. [151] conducted farm surveys in the major coffee growing areas of PR in 1998 and 2002 and concluded that the island was free of CBB. However, earlier false reports had significant consequences for PR’s coffee industry on the world trade market; for example, Mexico prohibited the importation of coffee products from PR [152].

The precarious situation of the Puerto Rican coffee industry was exacerbated when CBB was officially confirmed on the island in 2007 [153], after which time research institutions began to dedicate more effort to studying the insect and its impacts. In 2008, the first losses to the coffee industry resulting from reduced yield and quality of coffee due to CBB damage were quantified at more than US $15M. In 2009, the government implemented an adjustment table that would allow it to uniformly determine the damage caused by CBB, further reducing coffee profits [154]. By 2012, the number of coffee farms declined to 4478, with the total value of coffee grown in PR falling by US $12.6M [155].

CBB was initially confirmed in the municipalities of San Sebastian and Utuado [153], but according to an island-wide survey conducted in 2014 by Mariño et al. [139], CBB has become well-established throughout all coffee-growing regions of PR (Figure 2). Infestation and the total population per fruit were found to be positively correlated with elevation, with farms >400 m asl being heavily impacted [139]. Mariño et al. [112] reported that although infestation was higher in shade plots, coffee grown in the full sun had a higher female:male sex ratio and higher numbers of CBB per fruit. Shade coffee was also observed to harbor more natural enemies of CBB, including ants and entomopathogenic fungi [112]. While the use of other host plants has been reported in *H. hampei* [156], a recent study conducted in PR found that CBB rarely use fruits of alternate hosts, and did not feed or reproduce in these fruits [157].

An Integrated Pest Management plan for CBB was published by Gallardo-Covas and González [158] to assist in the control of this devastating pest in PR. This plan recommends four main actions: (1) sampling trees for evidence of infestation and monitoring flight activity using alcohol-baited traps, (2) the complete removal of all fruits from coffee trees at the end of the harvest season, (3) the application of *B. bassiana* based on CBB position in the fruit, and (4) sanitation and post-harvest control to prevent adult female CBB from escaping harvest bags during collection and transport and from facilities during processing. A recent study reported that ≈20% of coffee growers in PR were applying *B. bassiana* (generally only once per year), while post-harvest sanitation of tree and ground berries/raisins was not conducted due to high labor costs and low availability of workers [139]. Research is needed to determine the relative importance of each of these individual control measures, so that the most cost-effective plan may be recommended to growers.

Puerto Rico’s coffee industry has also been crippled by severe weather numerous times in its history. Most recently in September 2017, PR was hit by two major hurricanes: Hurricane Irma (Category 5) and Hurricane Maria (Category 4). Surveys from October–December 2017 found that almost half of the 81 plots examined had <20% damaged plants, whereas 12 plots across six municipalities had >80% of the coffee plants damaged [159]. Additionally, although many farms at low elevation finished harvesting for the season prior to the hurricanes, others at higher elevations reported 90% crop loss [159]. The high variation in damage and losses from these hurricanes meant that some farms managed to recover relatively quickly, while others needed to replant most of their crops or left the coffee industry altogether [159]. CBB populations were initially observed to decrease after the hurricanes, but this is likely because most fruits were harvested or destroyed [159]. Once conditions stabilized and coffee production resumed, their populations quickly recovered, and are likely to reach pre-hurricane levels as the coffee crop is re-established in the coming years [159]. With the frequency and intensity of natural phenomena predicted to increase (i.e., more extended periods of drought and extreme precipitation events, and more frequent tropical storms and stronger hurricanes [160]), climate change will likely continue to have adverse and severe effects on agricultural productivity at both local and global scales [161]. In addition to the direct destruction to coffee trees caused by major storms and drought, shifting precipitation patterns and temperature increases may result in the expanded distribution of crop pests and diseases [8,162].

Recent work in PR suggests the potential for future control efforts that are focused on the manipulation of bacteria found in the CBB microbiome. Mariño et al. [163] reported a diverse assemblage of bacteria in CBB that were dominated by *Pseudomonas* (associated with caffeine degradation [28]), while *Wolbachia* was the only endosymbiont detected. Bacterial communities were found to be more diverse in (1) adult females vs. eggs, (2) CBB collected from sun plots vs. shade plots, and (3) CBB collected from the field vs. CBB reared on artificial diet in the laboratory. The authors suggested that higher variation in CBB microbiota may be attributed to diet and exposure to a broad range of environmental conditions, particularly differences in daily temperatures and relative humidity [163]. The effect of eliminating *Wolbachia* from the CBB microbiome was examined by Mariño et al. [139] using artificial diet containing antibiotics. The authors reported that females fed antibiotics had significantly fewer progeny, lower fecundity, and fewer eggs per female. These results suggest that *Wolbachia* contribute to the reproductive success of CBB and that the manipulation of these endosymbiotic bacteria could act to suppress CBB population growth [139]. However, this approach needs to overcome the biological delivery challenges imposed by the cryptic behavioral reproduction of CBB.

### 7.2. Hawaii

Coffee is the fourth most valuable agricultural commodity in Hawaii, with production for the 2017–2018 season estimated at 24.6 million pounds (cherry basis) and an estimated value of $43.8 million [164]. Coffee arrived in Hawaii in 1817, with the first commercial operation planted in 1837 on the island of Kaua‘i [165]. Following a surge in coffee prices in 1975 and the decline of the sugarcane industry starting in the 1980s, the Hawaii coffee industry experienced new and expansive growth into 11 regions across the state [165]. *Coffea arabica* L. var. *typica* Cramer was introduced to Hawaii from Guatemala and is the most widely planted variety, although others such as Catuai, Caturra, Mokka, Bourbon, and Geisha are also grown. Hawaii coffee is cultivated under extremely variable climatic conditions from ≈200–800 m elevation on volcanic soils that vary greatly in terms of age and nutrient composition [166]. Almost daily cloud cover in the afternoons provides many farms above 370 m with natural shade during the hottest part of the day, such that most coffee in Hawaii is grown without shade trees [167]. A small percentage of farms have various fruit and nut species (*Persea americana*, *Mangifera indica*, *Musa* spp., *Citrus* spp., *Macadamia integrifolia*) or native/non-native tree species (*Metrosideros polymorpha*, *Acacia koa*, *Samanea saman*) inter-planted with coffee [167]. Hawai‘i Island is the center of coffee production in the state. The majority of coffee farms there are family-owned smallholder enterprises that are 0.5–1.5 ha [67]. The rough terrain and narrow rows on these small farms require that cherries be hand-picked, although there are large estates on Kaua‘i, O‘ahu, Maui, and Molokai that utilize mechanical harvesting.

CBB was first detected in the Kona coffee-growing district on Hawai‘i Island in August 2010 [168]. Following its initial detection, CBB spread rapidly to the ≈800 coffee farms on the island and was later confirmed on the neighboring islands of O‘ahu (2014), Maui (2016), Kaua‘i (2020), and Lanai (2020) (Figure 2). Infestation in unmanaged or poorly managed farms can exceed 90% [169]. Hamilton et al. [44] estimated that the number of CBB generations per season under natural field conditions in Hawaii was 2.1–3.3 for high-elevation farms (≈600–800 m) and 4.1–5.0 for low-elevation farms (≈200–300 m). The faster development rates reported for low-elevation farms is likely related to the narrow daily range of warm temperatures throughout the growing season [44].

Since 1888, a strict quarantine on coffee plants, plant parts, and used coffee bags entering Hawaii has been in place to protect the coffee industry. This law was amended in the 1970s to allow the importation of unroasted green coffee following either moist heat treatment or methyl bromide fumigation to meet demand for Kona coffee blends [170]. Given that coffee shipments are required by law to be fumigated before arrival in the islands, CBB was likely transported into the islands via infested berries that were accidentally carried in migrant worker’s clothing or luggage, although it is possible that CBB arrived in Hawaii in illegally imported or improperly fumigated shipments of green coffee [53]. Molecular phylogenetic and network analyses based on the COI gene region suggest that the most likely invasion pathway was from Kenya to Uganda to Latin America to Hawaii [53].

In the years following, detection, multiple programs, and government resources were directed at the problem of CBB in Hawaii, including ≈$5M spent to date by the United States Department of Agriculture (USDA) on a cooperative area-wide program; $1.55M by USDA and $750,000 by the Hawaii Department of Agriculture (HDOA) on subsidies and spray equipment for the entomopathogenic fungus *Beauveria bassiana* (commercially sold as BotaniGard^®^ ES and Mycotrol^®^ ESO); and additional funds for research from the state. Through the *Beauveria* subsidy program, growers were eligible to receive an annual reimbursement of 75% of the cost of this biopesticide (up to $600 per acre or $9000 per farm) from 2014–2016. The subsidy was extended by HDOA through June 2021, although the amount eligible for annual reimbursement has been lowered to 50% (up to $600 per acre or $6000 per farm). The HDOA also relaxed grading standards of premium Hawaiian coffees temporarily to assist growers as they dealt with learning to manage this new pest. Coffee is categorized and certified based on the number of defective beans per 300 g (≈1500 bean sample), and from 2014–2017, Hawaii Prime, Hawaii Natural Prime, and Hawaii Mixed Natural Prime were permitted to contain 20% defective beans by weight (the standard was reverted to a tolerance of 15% defective beans by weight on 1 July 2017). The governmental response can mostly be attributed to the highly organized nature of the Hawaii coffee industry, which includes multiple coffee grower associations. These organizations moved quickly to secure resources and external expertise to begin addressing the CBB problem in short order.

Although the programs mentioned above have mitigated the impacts of CBB on coffee production in Hawaii, the negative consequences of this invasion continue to be felt by growers, processors, buyers, and consumers. The estimated economy-wide impact of CBB for the crop years 2011/12 and 2012/13 in Hawaii was a $12.7M loss in crop value, a $25.7M loss in sales, a $7.6M loss in household earnings, and a loss of more than 380 jobs [171]. Prior to CBB arrival, coffee growers were required to spray fewer pesticides less frequently, and post-harvest sanitation was not necessary. Growers must now pay increased costs in order to produce the high-quality coffee that buyers and consumers have come to expect of the Hawaiian specialty brand. Greenwell Farms, one of the largest growers/processors in the state reported that before 2010 the long-term conversion of cherry to the green bean of grade Prime or higher was 5.49 pounds to one, compared to 8.89 to one in 2012/2013, and 6.9 to one in 2014/2015 [172]. Before the CBB invasion, 22% of the green beans processed at Greenwell Farms were graded as Extra Fancy and 30% as Fancy. In 2011 no coffee met the standard for either category; the company decided to implement measures to mitigate the devastating effects of CBB on the quality of coffee being processed at their facilities. As an incentive for growers to actively manage CBB on their farms, the company conducted quality testing for each bag of cherry delivered. A sliding scale was then used, in which the grower would receive the market price plus 10 cents per pound for cherry provided with 0–5% CBB damage (premium quality), and 10-cent deductions for each subsequent 5–10% increase in CBB damage (standard to inferior quality).

While this resulted in many growers taking their cherry to other buyers that were willing to pay top price for infested cherry, there were subsequent noticeable improvements in the quality of coffee brought to the mill, though well below pre-CBB yields: in 2015 11% of green beans were graded as Extra Fancy and 25% as Fancy [172]. Additionally, the rise in the percentage of coffee graded below Prime due to CBB damage (3% pre-CBB vs. 19–24% post-CBB) has driven up prices for certified green bean, particularly for the higher grades [172,173]. This has resulted in push-back from buyers across the US, Japan, and Europe who are unwilling to pay higher prices for lower grades of coffee and are beginning to look elsewhere on the world specialty market. The sale of Hawaiian coffee with high percentages of CBB damage continues to jeopardize the Hawaii coffee industry. In blind cupping trials, coffee sourced from Hawaii farms that had 5–15% CBB damage was detected by 17–50% of the cupping panel, while coffee with >20% damage was detected by 100% of the panel due to its negative impacts on aroma and flavor [174]. Thus, while consumers may initially be drawn to the brands’ exclusive reputation, they are unlikely to be return buyers of a product with low quality at a high price.

To improve the quality and production of Hawaiian-grown coffee, the University of Hawaii College of Tropical Agriculture and Human Resources (UH CTAHR) Cooperative Extension Service has conducted numerous outreach and education events (workshops, farm visits, conferences, information booths at festivals, etc.) aimed at helping coffee growers manage CBB and other coffee pests and diseases. The organization also hosts an educational website that provides access to CBB publications and presentations, demonstration videos for conducting monitoring and sanitation practices, information on recommended IPM practices, *Beauveria* subsidy program information and applications, and upcoming educational events and announcements. In 2013 and 2014 several organizations (UH CTAHR, HDOA, USDA-ARS, and various coffee grower associations) participated in a CBB Summit, intending to provide the coffee-growing industry with a set of recommendations for controlling CBB in Hawaii. The resulting document includes detailed information on each IPM component, including best practices for field sanitation, monitoring and sampling, pesticide application, harvesting, and shipping [102]. Annual surveys conducted by the UH CTAHR Cooperative Extension Service between 2012 and 2018 (*n* = 54–215 coffee growers) reported a 56% decrease in growers using traps for monitoring (20% in 2018), a 39% increase in growers using the 30-tree sampling method for monitoring (39% in 2018), a 17% increase in growers spraying *Beauveria* monthly (81% in 2018), a 22% increase in growers that were conducting strip-picking at the end of the harvest season (82% in 2018), and a 10% decrease in growers seeking and acquiring information on CBB IPM (56% in 2018) (A. Kawabata, pers. comm.).

### 7.3. Papua New Guinea

Coffee is the second-largest agricultural export in Papua New Guinea (PNG) accounting for 30% of the export earnings [175], with ≈87,000 ha producing 60,000 tons of coffee cherry annually [176]. The coffee industry is also the largest agricultural employer in PNG, providing the primary source of income to more than 2.5 million people (approximately half of the population of PNG) [177]. Coffee is grown in 15 of the 20 provinces, with about 400,000 smallholder farms (<1 ha), 660 larger farms (1–30 ha), and 65 commercial plantations (>30 ha) [177]. Smallholder farms produce 85% of the coffee grown in the country, with most farms located in the highlands where 70% of the population depends on subsistence farming [177].

PNG had the distinction of being one of only a few coffee-producing countries that were free of CBB until the pest was detected in smallholder farms in the Jiwaka, Eastern Highlands, and Morobe provinces in early 2017 [176]. A previous invasion in 2009 was quickly noticed and eradicated when it was located near the border with Indonesian Papua; however, the second incursion in 2017 appeared in an area that was far from the Indonesian border, and thus was not detected until it had spread to several provinces. Surveys by the Coffee Industry Corporation (CIC) and National Agricultural Quarantine and Inspection Authority (NAQIA) in early 2017 indicated that the pest was still confined to the areas where it was initially detected. Attempts to control its spread, including farmer awareness training and roadblocks in and out of the affected regions prohibiting the transport of cherry, seemed to be working initially but the beetle slowly continued to expand its range at a local level from farm to farm [178]. By February 2018, it was estimated that 17,486 ha were infested across four provinces (Figure 2), with mainly smallholder gardens being affected [179]. Consequently, roadblocks are being removed and NAQIA plans to cease involvement, with future priorities focusing on managing, rather than eradicating, the pest [179].

Management strategies currently recommended by the CIC and CIRAD (French Agricultural Research Centre for International Development) sustainable management services include farm sanitation, trapping, and biopesticide application [178]. Sanitation practices consist of complete removal of coffee berries from infested farms, although not all growers are participating in this measure and those that are find it difficult to be thorough. Rehabilitation of coffee farms (stripping, pruning, and spraying pesticides) was also encouraged by the CIC with the distribution of tools to growers [179], but many are not willing to lose income on the current crop to increase future yields. The only pesticide currently used to control CBB in PNG is chlorpyrifos, a highly toxic organophosphate. Research has been initiated to find safe and effective biocontrol agents, including *Beauveria bassiana*, of which at least one local isolate was successful at controlling CBB in field trials conducted in PNG in 2018 [180].

The potential impacts of CBB on the livelihood of growers and the PNG coffee industry are significant, being compounded by the already low yields resulting from rust, scale, and the old age of many coffee trees [179]. Production costs are expected to increase by 40% (to an estimated US$199 per ha) to keep CBB at manageable levels while still producing quality coffee [177]. PNG farmers generally have little awareness of pesticide safety or access to personal protection equipment and face increased risks of developing health issues due to the need to spray pesticides more frequently to control CBB. Additionally, the management of CBB is labor-intensive, and farmers are already experiencing labor shortages in PNG [181]. The Australian coffee industry is also at risk due to the CBB invasion in PNG, as it is close to the country (particularly North Queensland) and is currently free of CBB (Figure 2). Coffee berry borer is listed as the highest priority pest in the 2018 draft of the Australian Biosecurity Plan for the coffee industry, which is a relatively new industry with ≈520 ha of coffee and ≈155 growers [6].

### 7.4. Summary: Insights from Recent Invasions

The three recent cases highlighted here show that the initial invasion of CBB went undetected for long enough, or the response was not sufficient, such that attempts to curb the spread to adjacent coffee-growing districts or provinces were ultimately unsuccessful. CBB is capable of dispersing through both short and long-distance flight (either active or passive), and through human-mediated transport via infested coffee beans or in the clothing and harvesting equipment of workers. Perhaps one of the largest obstacles in successful eradication of a pest such as this is that many smallholder farmers are already on the brink of financial crisis, so complete removal and destruction of the crop early in the detection of the pest would result in many farmers being unable to recover. As such, the immediate response of eradication is often not feasible and coffee-growing regions are forced to focus on mitigation instead.

The most effective means of limiting CBB populations across all coffee-growing regions has been shown to be sanitation of the trees and ground, but most farmers are unable to find or afford labor to conduct this time-intensive cultural control. This may push farmers towards using less labor-intensive chemical controls, even if they know this management strategy is unsustainable in the long-term. Management strategies that limit the need for human labor are therefore a major need for the industry. Furthermore, a lack of knowledge of proper fertilizer use and ways to improve soil and tree health are common across all regions. With improved education and training in this area, farmers could greatly increase their yields and offset losses incurred from pests and diseases. In addition, with the frequency and severity of natural disasters expected to increase in the coming years, all coffee-growing regions stand to benefit from incorporating shade trees and windbreaks into their coffee farms. This single action would not only buffer coffee trees from the effects of intense drought, winds, and flooding but could also improve soil fertility and increase biodiversity of pollinators and natural enemies of CBB, including *B. bassiana*, nematodes, birds, and ants.

Lastly, lessons from Hawaii have shown that the involvement of coffee grower associations in the response and development of effective management strategies are crucial to the success of any IPM program, as they are able to best represent the interests and needs of smallholder farmers to ensure that any proposed strategy is cost-effective and feasible. The cooperation of government, industry, researchers, and coffee growers was essential in laying the groundwork for a sustainable management program in Hawaii, which we discuss in further detail in the following section.

## 8. Current State of CBB Management: Hawaii as a Case Study

In this section we describe the major components of CBB IPM strategies currently used in most coffee-growing regions around the world, highlighting some of the most recent research conducted in Hawaii. We also describe challenges that need to be addressed to improve CBB management within an IPM framework. Given that the research on CBB in Hawaii is ongoing and the literature is continuously expanding, this section is not meant to be an exhaustive review, rather we aim to provide an overview of major efforts to control this serious pest.

### 8.1. Monitoring: Traps and Tree Sampling

Funnel traps are used in many coffee-growing countries to monitor the flight activity of adult female CBB (Figure 5A). Peaks in flight activity can inform optimal spray times, as this is when the CBB are exposed and most vulnerable to insecticides [64,111,182]. Traps are baited with an alcohol lure that mimics kairomones released by developing coffee berries [183]. The response of CBB to different lure formulations was examined at a single farm in Kona by Messing [184], with results suggesting that 3:1 and 1:1 combinations of methanol:ethanol performed equally well at attracting CBB to traps, while 100% isopropyl alcohol attracted significantly fewer beetles. A study conducted at 15 farms on Hawai‘i Island by Aristizábal et al. [185] showed that trap catches correlated well with berry infestation at the majority of farms examined, particularly early in the season. This suggests that traps could be used by growers to inform the timing of sprays early in the season. Although several studies have attempted to develop action thresholds based on trap catches, this has proven difficult due to the notoriously heterogenous distribution of CBB in fields and its complex relationship with coffee phenology and environmental variables [111,185,186]. Further research is also necessary to determine if mass-trapping (22–25 traps/ha) [64] can be used as part of an IPM program to reduce CBB populations remaining on farms in the inter-crop season.

In 2012, the 30-tree sampling method for CBB monitoring developed in Colombia by Centro Nacional de Investigaciones de Café [30,187] was introduced to coffee growers in the Kona and Ka‘u districts on Hawai‘i Island. This method allows growers to determine CBB infestation percentage, CBB position within the berry, and the locations of hotspots of activity within the farm [188]. Trees are randomly sampled in a zig-zag pattern across the farm (30 trees per 2.5-acres or 12 trees per 1-acre), and from each tree, a single branch is selected [68] (Figure 5B). Percent infestation is calculated by dividing the number of infested green berries on the branch by the total number of green berries on the branch, and then multiplying by 100. Three infested berries from each branch are dissected to determine the percentage of CBB in the AB position that is alive (important for making spray decisions) and the portion that is in the CD position (important for determining bean damage). Although this sampling method provides an accurate estimation of CBB infestation and positions, and is an ideal way to sample farms for scientific studies [189], it is labor-intensive and time consuming for growers to use regularly.

Fixed-precision sequential sampling has recently been proposed as a more efficient method of estimating CBB density in Hawaii coffee farms. Using this method, sampling is terminated when the estimation of pest density is reached with the desired precision level [190,191]. Aristizábal et al. [192] found that to detect 1.5–2.5% infestation (suggested economic threshold) at a fixed precision of 10–25%, 6–50 coffee branches/ha would need to be sampled. This study also found that modifying the protocol to count only the infested green berries (vs. the total number of green berries) on each branch required 27% less time, and that the results of the two protocols were highly correlated. Pulakkatu-thodi et al. [193] also used fixed-precision sequential sampling to estimate CBB density on 12 Hawai‘i Island coffee farms and compared berry clusters and branches as sample units. The authors found that sequential sampling of berry clusters required detection of fewer infested berries relative to branch sampling, although both were reliable methods for estimating pest density. These data are important as they suggest that the fixed-precision sampling method is more efficient compared to the 30-tree sampling method and could be a quicker alternative for growers to assess CBB activity in their fields. However, recent surveys conducted by CTAHR indicate that adoption of field sampling by coffee growers in Hawaii has decreased over the last two years (high of 77% in 2016 compared to 39% in 2018), with 73% of growers preferring to visually monitor their fields (looking for presence of CBB entrance holes in berries, as opposed to using traps or the 30-tree sampling method; A. Kawabata, pers. comm.), a result that is likely directly related to the time required to conduct sampling.

### 8.2. Cultural Control: Pre- and Post-Harvest Sanitation

Farm sanitation is the most important component of CBB IPM [67,68,102,188]. Prior to the arrival of CBB in 2010, sanitation practices were not imperative for Hawaii coffee growers. Traditional harvesting was conducted at the end of the season, with little concern for the number of cherries left on the trees or ground. Johnson and Fortna et al. [194] examined six farms on Hawai‘i Island during the inter-crop season to quantify post-harvest population reservoirs of CBB. They found that after strip-picking, 49.5% of the total CBB load was present in raisins left on the trees, 47.3% was in raisins in the dripline (ground below the tree foliage), and 3.2% was in raisins in the center aisle (ground between rows). The authors also found a significant positive correlation between ground raisin density and percent infestation in the following year’s crop. This research suggests that tree raisins are the main reservoirs for CBB during the inter-crop season, and ground raisins are a significant secondary source. To minimize CBB in tree raisins, strip-picking is now recommended at the end of the main harvest season (Figure 5C). This practice removes all green, red, over-ripe, and raisin berries from the trees, and must be conducted prior to pruning to limit CBB dispersion. Coffee workers may be paid hourly or by weight to perform strip-picking. Sanitation picking, in which cherries and raisins are picked every 2–3 weeks, may be conducted as an alternative to strip-picking in regions that have a year-round growing season (e.g., Ka‘u district on Hawai‘i Island). Evaluations conducted by the Synergistic Hawaii Agricultural Center (SHAC) on 11 Hawai‘i Island farms across 36 standard harvesting rounds found that good to excellent harvesting efficiency (<10 mature berries remaining per tree) corresponded to low CBB infestation levels (<2%), while bad harvesting efficiency (>10 ripe berries per tree) corresponded to infestation levels at or above the economic threshold (>5%) [67].

Ground raisin management is not currently implemented by most growers in Hawaii, mainly due to high labor costs (roughly $2/lb or $10/hr; L. Aristizábal and J. Ah San, pers. comm.) and rough/rocky terrain that makes ground raisin removal difficult. However, several options may be feasible for limiting ground raisins, including picking on tarps to minimize the number of cherries dropped on the ground, increasing the frequency of traditional harvesting, implementing the use of mechanical collectors to remove raisins from the ground, planting ground covers that could assist in raisin decomposition and support natural enemies of CBB, or spraying *B. bassiana* or entomopathogenic nematodes on ground raisins [194]. Further research is needed to explore the cost–benefit and feasibility of each of these methods for particular coffee-growing areas.

Post-harvest control of CBB can be achieved by implementing several measures in the field and processing facility. Burlap cherry sacks should be lined with industrial plastic bags to prevent CBB from escaping back into the area, and full bags should be brought to the wet mill for processing as quickly as possible [195]. Used burlap sacks can be soaked in hot water for 15 min to kill any remaining CBB. Beetles that survive during the pulping process may be trapped by placing transparent plastic on the floater lid and smearing with grease; this may also be done for the fermentation tank. Alcohol-based traps should be placed around the wet and dry mill to capture any CBB that escape from the processing facility. Small quantities (<15 lbs.) of infested berries collected during strip-picking or sanitation picking may be boiled for 30 min [196] or frozen for 48 h to kill CBB and prevent re-infestation [197]. Larger quantities may be placed in a dryer machine for 60 min at 131 °F [169], or placed in industrial plastic bags or sealed buckets and left in the sun for 2–3 weeks [195]. Collected berries may also be buried at least one foot below the ground surface [195].

Stump pruning by block is currently recommended in Hawaii as an effective way to reduce CBB populations over large areas [102]. This method differs from the traditional Kona-style (multiple verticals of a different age on the same tree) and Beaumont–Fukunaga style of pruning (all verticals are the same age on a tree but differ across rows) in that all the trees in a block are stump pruned, ensuring that sources of food and shelter for the pest are removed. Aristizábal et al. [67] examined three adjacent farms on Hawai‘i Island one-year post-pruning and reported that while infestation increased during harvest for all three farms, it remained lowest in the farm that employed stump pruning by block (3.7% infestation in stump-pruned farm vs. 6.8–12.0% infestation in Beaumont-Fukunaga pruned farms), and had lower total defects in dried green coffee beans (3.9% vs. 10.2–10.6%). Stump pruning should be conducted after strip-picking (except in periods of drought), and also in fields that are not actively being harvested to reduce sources of infestation for neighboring farms. Research is currently being conducted on Hawai‘i Island to examine the interaction of various pruning techniques with other IPM components. For example, plant growth regulators applied to coffee trees to synchronize flowering and berry development have been found to be most effective on trees that are stump pruned [198]. Removing suckers and pruning branches from coffee trees with dense foliage can also help to reduce infestation by allowing better spray coverage and providing easier access to berry clusters during harvesting.

### 8.3. Biological Control: Natural Enemies

The entomopathogenic fungus *Beauveria bassiana* is one of the most important natural enemies of CBB worldwide, with *B. bassiana*-induced mortality levels reported to range anywhere from <1–70% in field populations [5]. In areas that have ideal environmental conditions for fungal growth (moderate temperature, high humidity, low insolation) [199,200], infected adult CBB females may be seen with a fluffy white mycelial outgrowth extending from the posterior end of the beetle (Figure 5D) that is exposed when boring activity is initiated (AB position) [201]. The GHA strain of *B. bassiana* (sold commercially as BotaniGard^®^ ES and Mycotrol^®^ ESO) has been an important component of IPM programs in Hawaii since it was first registered for use on coffee by HDOA in February 2011. The majority of growers surveyed for the 2018–2019 season sprayed frequent (anywhere from 3–8 sprays per year; M. Bondera, pers. comm.) applications of this biopesticide to control CBB on their farms, although the timing, dosage, and interaction with other management practices are seldom taken into consideration.

Greco et al. [202] reported that the effectiveness of *B. bassiana* applications was influenced by the position of the female in the berry, with higher mortality in the A and B positions relative to the C position at three farms on Hawai‘i Island. BotaniGard^®^ ES applications also appeared to be more effective with an increase in elevation [202], with high-elevation farms on Hawai‘i Island experiencing afternoon cloud cover for much of the coffee-growing season. Wraight et al. [201] also reported that the highest activity of wild-type (feral) strains of *B. bassiana* on Hawai‘i Island was at sites >500 m in elevation, with 24–42% of foundress CBB in green berries infected. In contrast, infection rates did not exceed 4% at low-elevation (<300 m) farms on Hawai‘i Island [201].

A recent study examining calendar (≈monthly) vs. threshold (based on field monitoring data) spray strategies of *B. bassiana* found that farms adopting the threshold strategy applied *B. bassiana* early in the season (May–July) and required fewer sprays to achieve the same level of control achieved with calendar sprays, resulting in a net benefit [203]. Research on the effect of dosage suggests that reduced rates (16 oz per acre) of BotaniGard^®^ ES may be just as effective as the recommended rate (32 oz per acre) in controlling CBB, particularly when combined with acceptable sanitation practices (L. Keith and S. Wraight, pers. comm.). A decision analysis conducted for CBB in Hawaii found that the initial infestation level was the most critical factor in maximizing the final net benefit of spraying *B. bassiana* [204]. Good pre- and early-season sanitation practices (e.g., strip-picking, pruning) work to keep infestation levels low throughout the growing season. If infestation levels are high at the beginning of the season, spraying *B. bassiana* will do little to reverse crop damage [102,204].

The collective results of these studies provide several guidelines for improving the effectiveness of *B. bassiana* as part of a successful IPM program for CBB. First, farm sanitation practices including strip-picking (removing all berries from the trees at the end of the main harvest season), sanitation picking (removing raisins and cherries every 2–3 weeks), pruning of suckers and dense foliage, and ground sanitation (raisin removal) must be conducted to ensure that initial infestation levels are low. Second, the ideal time to spray *B. bassiana* is early in the coffee-growing season, as it is at this time that the majority of CBB females are in the AB position and are most likely to come in direct contact with *B. bassiana* conidia [67,185,189]. Third, the timing of applications both in terms of the local weather conditions and the flight activity of the beetle must be considered. Applications done late in the day (CBB tend to swarm and attack the crop between 2 and 6 PM; I. Pulakkatu-thodi, pers. comm.) when the weather is cloudy and humid but not raining (optimal conditions for spore persistence) are likely to be most effective [201]. Fourth, the locations of farms must be considered in decisions of whether or not to spray, as farms that may experience weather conditions that will reduce spore viability (high solar radiation, high temperature, and low humidity) for much of the year will have fewer opportunities to use this biopesticide. The planting of shade trees to make conditions more favorable to fungal survival, or an increased emphasis on sanitation instead of spraying this biopesticide, may be a viable solution for such a farm. Conversely, regions with high precipitation may need to use surfactants to prevent fungal spores from washing away [195]. Lastly, vigorous mixing must be done to ensure spores are adequately suspended, and the proper spray equipment must be determined depending on the size of the farm (e.g., backpack or mist sprayer for small farms, tractor with mist sprayer for large farms) [195].

Adult flat bark beetles, particularly *Leptophloeus* sp. and *Cathartus quadricollis* (the square-necked grain beetle), are natural enemies of CBB that have been observed inside infested coffee berries in Hawaii. Molecular markers confirmed that these flat bark beetles were feeding on CBB in Hawaii coffee farms [205], and laboratory assays suggest that these predators will feed mainly on immature CBB life stages [206]. Flat bark beetles are widely distributed across the coffee-growing districts of Hawaii, but have mostly been observed feeding in tree raisins [206]. This may limit the extent that these natural predators can be used to control CBB since crop damage occurs in the green and mature berries. Nonetheless, flat bark beetles can play an important role in managing CBB populations in tree raisins [207]. Importantly, these beetles were found not to be susceptible to *B. bassiana* infection and can easily be raised on a diet of cracked corn and cornmeal [206]. The flat bark beetle project funded by HDOA and in collaboration with UH CTAHR and USDA-ARS offers coffee growers starter kits with a beginning population of 50–100 *C. quadricollis* that can be used to rear large quantities of the beetle for augmentation of natural field populations. Recent work has focused on the development of predator breeding stations for augmentative biological control. A *Cathartus quadricollis* aggregation pheromone is used to attract wild beetles to food suspended in a bucket in trees where the beetles reproduce and multiply and then disperse back into the coffee field. A single breeding station attracting 100 *C. quadricolis* can produce 10,000 beetles over a 4-month period, which disperse from the station and can predate on CBB in coffee fields (P. Follett, unpub. data). The *Cathartus* breeding stations and lures are available commercially through Alpha Scents (Portland, OR).

Entomopathogenic nematodes (EPNs) are natural enemies of CBB that use mutualistic bacteria to infect and kill the beetle, followed by feeding and reproduction in the cadaver. EPNs have the potential to be developed as a commercial biopesticide that could be sprayed on tree berries and ground raisins [208]. Nematodes in the genera *Heterorhabditis* and *Steinernema* have been tested against CBB in laboratory assays, with variable mortality rates [209,210,211]. Importation of EPNs is currently prohibited in Hawaii, except for *Steinernema carpocapsae*. Trials conducted on Hawai‘i Island using CBB-infested berries that were placed on soil and sprayed with *S. carpocapsae* showed that infective juveniles of the nematode were capable of entering the berries and killing CBB in all life stages [212]. Average nematode-induced mortalities of 26.6% for adults and 23.7% for larvae were observed in laboratory trials, and average mortalities of 4.7% for adults and 17.1% for larvae were observed in field trials [212]. Two other species of EPNs (*S. feltiae* and *H. indica*) are already present in Hawaii, and therefore do not require permits for testing. Trials conducted on ground raisins on Hawai‘i Island with these two species using sprays of infective juveniles and nematode-infected mealworm cadavers showed that inoculation did not cause significant CBB mortality but did result in abandonment of the berries (R. Meyers and R. Hollingsworth, pers. comm.). Further research is needed to determine the most effective way to use EPNs in their natural environment (near the soil surface) as part of an IPM program for CBB. One potential method would be to combine nematode releases on the ground with appropriately timed *B. bassiana* sprays that could then kill CBB as they abandoned ground berries. Nematode activity is promoted by moist soil surfaces and high humidity, which may explain the higher nematode-induced mortality achieved in greenhouses compared to field environments [213]. Additional work is needed to optimize methods of keeping nematodes alive and above-ground for long enough to initiate entry into CBB-infested ground berries.

Explorations for natural enemies of CBB in Africa have revealed several parasitoid wasps that may be useful as biocontrols for this pest. *Cephalonomia stephanoderis* and *Prorops nasuta* are larval-pupal ectoparasitoids of CBB that usually prey on females, eggs, and first instar larvae [214,215], and have been introduced as a biocontrol in the Americas. Although these parasitoids have been successfully established in all the countries where they were released, parasitism levels quickly drop-off after the first month, and were rarely detected after 12 months [5]. This suggests that multiple releases would need to be conducted in a season to reduce CBB populations below the economic threshold. *Phymastichus coffea* is an endoparasitoid that attacks CBB adults and has been released in at least 12 countries. Females oviposit in the abdomen of the CBB adults, laying a single male and a single female egg, which hatch and feed on the internal tissues of the host. Host CBB that are parasitized by *P. coffea* dies within 15 days [216]. This species of parasitoid is considered ideal for use as a biocontrol agent against CBB. Although it is short-lived (2–3 days), it can be released any time after fruit colonization, with studies showing successful parasitization up to seven days after CBB have initiated berry entry [216,217]. Host testing in quarantine by USDA-ARS with native and exotic scolytines and other Coleoptera suggests this parasitoid is highly host specific (F. Yousuf and P. Follet, pers. comm.). Efforts are underway to obtain the necessary permits to import this parasitoid into Hawaii for inundative releases.

### 8.4. Chemical Control: Insecticides and Repellants

Several coffee-approved insecticides have been laboratory tested for efficacy against CBB in Hawaii, with mixed results. Insecticides containing imidacloprid (Provado^®^), potassium salts of fatty acids (M-Pede^®^), spirotetramat (Movento^®^), clarified hydrophobic extract of neem oil (Trilogy^®^), azadirachtin (Neemix^®^ 4.5), or sodium tetraborohydrate decahydrate (Prev-Am Ultra) as the active ingredient showed no effect on adult female CBB with direct or indirect contact [102]. In contrast, the pyrethrin-based insecticide EverGreen^®^ Crop Protection EC 60-6 exhibited control when directly applied to female CBB adults [102]. CBB repellants that have been tested on coffee in the field include Garlic Barrier^®^ AG+ (garlic extract-based), Surround^®^ WP (kaolin clay-based), and Captiva^®^ (capsicum extract-based). While these products show some promise for repellancy and direct control of CBB both alone and in combination with BotaniGard^®^ ES, good coverage would need be achieved and the product frequently reapplied, particularly during periods of heavy rainfall [102,195]. New products for chemical control or repellence of CBB face development and screening, regulatory, and economic (limited market size in Hawaii) hurdles.

### 8.5. Physical Control: Exclusion Netting and Border Crops

Physically controlling arthropod pests is one of the oldest IPM techniques and simultaneously a potential avenue for future innovation. While physical control has not been a component of CBB IPM programs to date, two studies conducted on Hawai‘i Island suggest that physical barriers to CBB flight may be a feasible option for controlling CBB, particularly on smallholder farms. In an investigation of CBB abundance in coffee sites with varying management intensity, Johnson and Manoukis [169] found significantly reduced trap catch and infestation in feral and abandoned coffee sites compared to well-managed and poorly managed sites. The authors suggested that the CBB movement in these feral and abandoned sites may be inhibited by the presence of dense vegetation surrounding coffee plants. Border crops that act as physical barriers to CBB flight between coffee plots may therefore be an option for reducing movement of this pest in areas with high connectivity and high density of coffee plants (e.g., Kona district on Hawaii Island). An alternative to natural barriers is fine-mesh exclusion netting, which is made from high-density polyethylene (HDPE) material that can last under field conditions for 5–7 years. Exclusion nets prevent insect pests from accessing plants in addition to modifying the growing environment (e.g., reducing solar radiation, increasing relative humidity), and have become widely integrated into the production of many commodity crops. A recent study by Johnson et al. [135] examined exclusion netting for the control of CBB on two coffee farms located on Hawai‘i Island and found that netting significantly reduced CBB infestation in green and ripe coffee. Additionally, this study reported no negative impacts on coffee yield or quality, suggesting that exclusion netting may be a viable future alternative to chemical controls.

## 9. Biosecurity Issues

The term “biosecurity” has increased in usage significantly since the start of the 21st century, but the term can have multiple meanings “on the ground,” politically, and scientifically [218]. Further, this term has been applied to diverse situations from agriculture, to marine area protection, to disease management, always with variations in meaning depending on the context [219,220,221,222]. A useful functional definition is that “biosecurity is the exclusion, eradication or effective management of risks posed by pests and diseases to the economy, environment and human health” [223]. A review of the literature suggests that this formulation of biosecurity as management of risk resulted from the confluence of ideas on biological invasions with heightened attention to security matters such as bioterrorism at the turn of the century [222,224]. Some of the best current biosecurity programs are in New Zealand and Australia, the former being the only country in the world with specific biosecurity legislation, where the integration of components is necessary to minimize risk. This is sometimes termed the biosecurity “continuum,” reflecting the chain of events needed for a successful biological invasion (incursion, spread, and establishment) and the attendant security controls on each step (exclusion, surveillance, and eradication/control). It is important to stress that agricultural practices that today might be termed “biosecurity” have existed for decades or centuries on farms [218], but the combination of these with components from the other parts of the continuum can lead to risk management and what is understood to be a valid biosecurity program in the contemporary sense.

Coffee can be the vehicle of significant biosecurity issues, mainly because global trade is often cited as a key driver of biosecurity problems in agriculture [225]. Coffee is the second most traded good in the world (petroleum is first), with *Coffea arabica* and *Coffea canephora* making up the majority of the coffee trade [226]. Coffee exports in 2009–2010 totaled US$15.4 billion, with 93.4 million bags of coffee shipped throughout the world [227]. Over 100 million people depend on coffee for their livelihood [227]. Movement of unprocessed coffee and coffee workers sets the stage for a large number of incursions by CBB and will likely enable other coffee pests and pathogens to enter new areas. One lesson from the world’s experience with CBB is that a comprehensive approach is needed to minimize the biosecurity risk of future coffee pests and diseases.

A comprehensive biosecurity approach would include at least four major components: (1) post-harvest treatment of any coffee exported from the area of production; (2) pathway analysis and port of entry monitoring to allow pest and pathogen exclusion; (3) monitoring and surveillance of port and growing areas; and (4) a rapid emergency response plan for effectively responding to an incursion. The approach outlined here is not specific to coffee, and indeed could be applied to a range of agricultural commodities and pests. In the case of coffee, the stability of the processed end-product affords a highly effective and straightforward method for preventing the movement of pests and pathogens from the country of origin to destination: processing and roasting at the origin. Biosecurity resources of destination markets could thus effectively be employed on technology transfer and creating infrastructure to allow vertical integration in producing regions. This has economic consequences for the value chain of the coffee trade, as it would enable the producing regions to bring to market a higher-value commodity than unprocessed cherries; this might bring economic development to producing regions, likely a net benefit in multiple areas. In any event, these considerations would have to be taken together with the advantage of reducing pest and pathogen movement.

While it is critical for the remaining coffee-producing regions that are CBB-free (Nepal, Australia, China) to consider the four biosecurity components described above to protect their coffee from CBB, restricting the movement of this pest in areas where it is already established might also be necessary. Reasons include the possibility of introducing associated pathogens that might not yet exist at these locations or are worse than the local lineages or races (e.g., the fungus associated with coffee berry disease which may be spread by CBB) [228]. For areas that already have CBB but want to minimize the introduction of new populations or novel pests and diseases, the focus should be on ensuring effective phytosanitary treatment of any imported material, followed by exclusion via various pathways and surveillance/response planning.

Research on post-harvest treatments for CBB has been conducted in recent years [197,229,230], but the question of “who pays” is most often the sticking point to seeing these methods applied. The administrative and economic arrangements needed for specific instances are beyond the scope of this paper, but it is worth noting that this type of treatment is the most cost-effective way of preventing the movement of a pest such as CBB into a new area. It is possible that such a program could have prevented the introduction of CBB to the three island regions examined here, as these have few and well-defined ports of entry. Surveillance for pest incursions should include monitoring ports of entry with traps, cargo inspections, and public education campaigns. A more recent idea is to employ the public as surveillance components, a feature of “Biosecurity 2025” in New Zealand. Such an effort must include outreach and social science components, and surveys of attitudes [231]. This component is typically more expensive than post-harvest treatments, but costs are usually borne by governmental agencies rather than directly by producers. Once CBB has entered an area, an effective response plan would have to include the destruction of significant numbers of coffee plants (F.E. Vega, pers. comm.). The extent of the area that would have to be eliminated depends on how soon the incursion is detected. Funds to compensate growers and possibly emergency special use pesticide permission could be established in the months and years ahead of a detection to facilitate timely response and increase the odds of successful eradication. This sort of program is the most expensive of all to operate.

## 10. Conclusions

Several parallels can be drawn from historical and recent invasions of CBB throughout the coffee growing countries of the world. Once introduced to a given region, the spread of CBB was generally rapid and very difficult to contain, with all commercially grown species and varieties of coffee being affected across a wide range of elevations. Socioeconomic issues, including high labor and production costs, low labor availability, and a lack of worker education and training in CBB management, have greatly inhibited the implementation of successful IPM programs in all coffee-growing countries. Additionally, the historical reliance on chemical controls that negatively impact human and environmental health and can lead to insect resistance have greatly impeded the adoption of more sustainable management strategies. Lastly, there is evidence from multiple countries that increasing temperatures are resulting in shortened CBB development times, such that coffee growing regions predicted to experience rising temperatures under future climate change scenarios may be subjected to more intense pressures from this pest.

Across all coffee-growing countries, sanitation has been recognized as the most essential component of any CBB IPM program. The removal of all remaining tree and ground berries/raisins after the harvest season is complete has long been a vital management technique employed in Latin American coffee-growing countries, made possible by the low cost of labor and high availability of workers. For regions where this is much more difficult due to labor constraints and high costs, and for areas with year-round growing conditions, additional improvements could be made with more efficient and frequent harvesting practices. Worker education and training are essential for adequate farm sanitation, and along with better tools (e.g., the use of wider picking baskets and tarps to limit the number of cherries dropped, mechanical collectors to quickly pick-up ground raisins) and harvesting methods (e.g., picking cherry and raisins every 2–3 weeks), this single IPM component can be highly effective at managing CBB populations and limiting infestation in future crops. Keeping trees pruned to a manageable height and weed suppression are also important sanitation aspects that can significantly improve the ability of workers to harvest the coffee efficiently. Supplementing good farm sanitation practices with threshold-based applications of biopesticides (e.g., *Beauveria bassiana*) and the release of host-specific biocontrols (e.g., parasitoid wasps) will likely be sufficient for keeping CBB infestation below economic injury levels.

Lastly, in addition to the topics of future research discussed in this review, we suggest that there is great potential for improvement of CBB IPM programs by incorporating technological advances in precision agriculture. For example, ground-based remote sensing techniques may be used to detect increases in flight activity and areas of high pest density, though sensitivity improvements are needed [232]. Mobile application technology also stands to improve monitoring efficiency and coupled with networks of sensors collecting information on local weather conditions, can be used as a decision support tool to determine the appropriate management actions for specific locations. Knowledge of CBB biology, local weather patterns, and electronic data collection via mobile applications can also be used to build predictive models of CBB flight activity and infestation, allowing further tailoring of IPM strategies to specific growing locations. These technologies might be combined with automated mechanical systems for post-harvest sanitation or even automated trapping during the inter-crop season. Together, these technological advances along with increased knowledge of CBB biology and the connection of their life cycle to specific weather variables hold great promise in the development of a sustainable and successful IPM program for this global pest of coffee.

## Figures and Tables

**Figure 1 insects-11-00882-f001:**
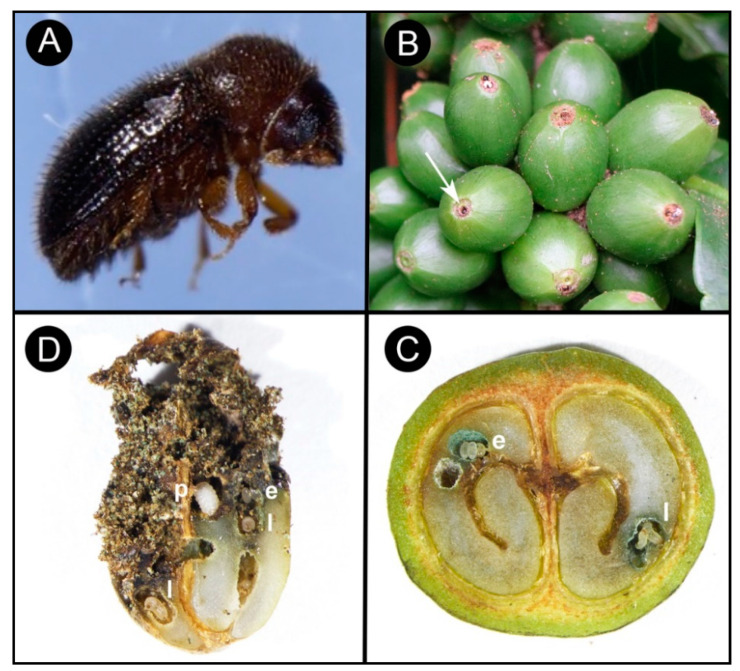
The coffee berry borer life cycle: (**A**) adult female coffee berry borer; (**B**) entrance hole with female coffee berry borer (CBB) (arrow) in the central disc of the developing green coffee berries; also note the presence of entomopathogenic fungus *Beauveria bassiana* (white mycelium protruding from female CBB); (**C**) reproductive galleries containing eggs (e) and larvae (l) in the coffee bean; (**D**) advanced stages of bean damage due to larval feeding, with pupae (p), larvae (l), and eggs (e) visible.

**Figure 2 insects-11-00882-f002:**
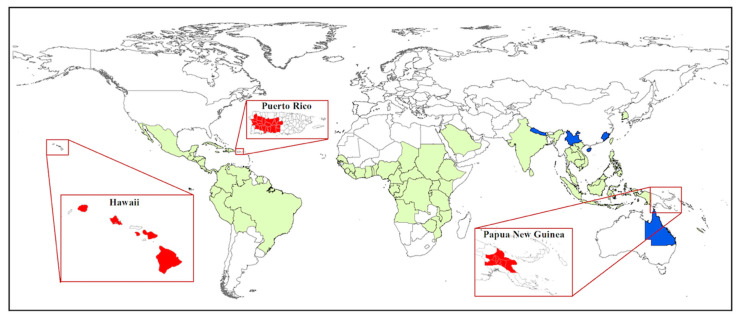
World distribution of the coffee berry borer (CBB). Countries with CBB prior to 2007 are highlighted in green. Island regions that were recently invaded are highlighted in red from left to right; Hawaii (2010), Puerto Rico (2007), and Papua New Guinea (2018). The only countries without CBB (Nepal, China, and Australia) are highlighted in blue.

**Figure 3 insects-11-00882-f003:**
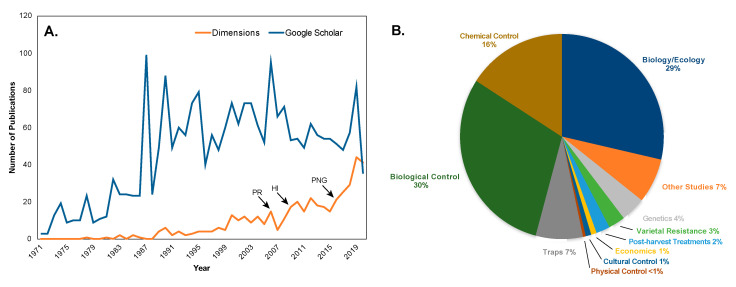
Number of publications per year on *Hypothenemus hampei* (CBB) over the last 50 years as reported in the Dimensions and Google Scholar databases (**A**). The arrows indicate when CBB was introduced to Puerto Rico (PR), Hawaii (HI), and Papua New Guinea (PNG). Over the last 10 years (2011–2020), the majority of publications have been focused on describing the basic biology and ecology of CBB, and investigating biological and chemical controls (**B**).

**Figure 4 insects-11-00882-f004:**
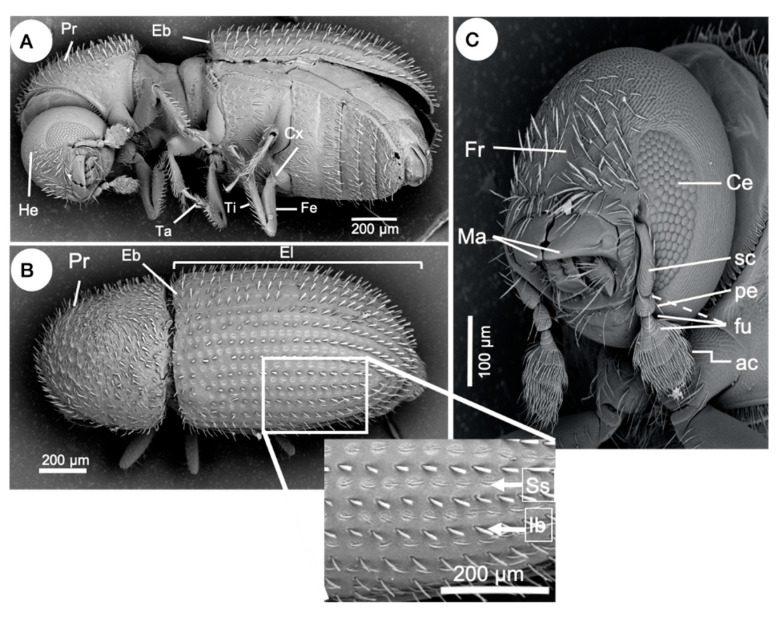
External anatomy of an adult of *Hypothenemus hampei.* (**A**) Lateral and inferior view showing pronotum (Pr), elytral base (Eb), head (He), legs with third coxa (Cx), femur (Fe), tibia (Ti), and tarsus (Ta). (**B**) Top view; detail (box) of the elytron (El) showing strial setae (Ss) and interstrial bristles (Ib). (**C**) Frons (Fr), mandibles (M), compound eye (Ce), antenna club (Ac), segmented funicle (Sf), pedicel (Pe), scapus or scape (Sc). Nomenclature according to Hulcr et al. [20] and Vega et al. [5].

**Figure 5 insects-11-00882-f005:**
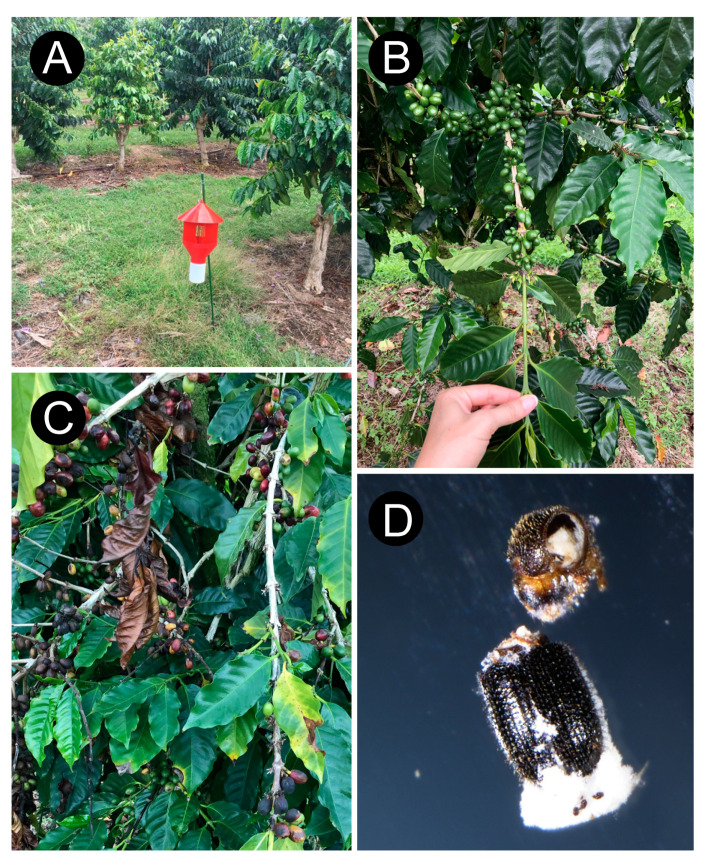
Essential components of a successful IPM program for CBB: monitoring flight activity using funnel traps baited with a 3:1 methanol:ethanol attractant (**A**); infestation assessment using the 30-tree sampling method (**B**); post-harvest sanitation of all berries, including over-ripe and raisin berries (**C**); and early season sprays of *Beauveria bassiana* fungus (**D**).

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
