# Peer review of "Coffee Berry Borer (Hypothenemus hampei), a Global Pest of Coffee: Perspectives from Historical and Recent Invasions, and Future Priorities"

_insects, 2020, doi:10.3390/insects11120882_

Round 1

Reviewer 1 Report

Manuscript review

Coffee berry borer (Hypothenemus hampei), a global pest of coffee: perspectives from historical and recent invasions, and future priorities

Johnson et al.

Some comments and observations on the manuscript that are intended to help improve it are the following:

Line 18 and line 37: It says Ferrari, it must say (Ferrari).

Lines 40-41: In order to match the photograph (Fig. 1 B), it should be clarified that the insect bores the fruit (berry) of the coffee to later enter the seed (bean).

Line 40: Apparently CBB has not been reported in China either. Check it out.

Lines 74-75: It is not clear how these figures were obtained (in one of the countries mentioned there are more publications than those mentioned). Therefore, more detail is required on how the selection of the publications was made. It is desirable that the authors include the list of the mentioned publications as supplementary material.

Line 75: In Fig. 3, indicate with an arrow (or other symbol) the year of the introduction of CBB to US coffee-producing regions.

Line 92: It is not clear which is the reference from which the data was obtained for 181 described species of the genus Hypothenemus.

Line 95: I didn’t find Waterhouse and Norris (1989) in the References section.

Line 116: I didn’t find Hulcr et al. (2015) in the References section.

Line 118: Cite the reference that the CBB was first reported in Liberia in 1897.

Line 127: Probably the maximum altitude for robusta (C. canephora) in West and Central Africa of the mentioned range (250-1500 m), is incorrect. Check it out.

Line 134: It is imprecise to say “when the dry matter content is 20% or greater”, for example, see Fig. 7, p. 74 in Barrera (1994) [Barrera, J.F. 1994. Dynamique des populations du scolyte des fruits du caféier, Hypothenemus hampei (Coleoptera: Scolytidae), et lutte biologique avec le parasitoide Cephalonomia stephanoderis (Hymenoptera: Bethylidae), au Chiapas, Mexique Ph.D. thesis. Université Paul Sabatier Toulouse, France, 301 pp] (http://www2.tap-ecosur.edu.mx/mip/TesisJFB/12_Chapitre_III_Floraison.pdf). This figure shows that before reaching 20% dry weight, when the coffee seed is not suitable for CBB to initiate oviposition, the dry weight is also greater than 20%, however, at this stage the endosperm of the seed is not yet has completed its development. According to this figure, it is more appropriate to say “when the dry weight content of the coffee seed is 20% and the endosperm is in the state of development known as semi-consistent”. This is true for the case of C. arabica.

Line 136: In many cases, CBB also bored next to the floral disc, not only in its central part.

Lines: 137-138: “…unless the infestation is high…” and the availability of berries is scarce.

Line 165: Same as Line 40.

Lines 147-148: I suggest mentioning the mode of sex determination of CBB, for example the “functional haplodiploidy” and discussing its importance in CBB colonization of new regions (Brun et al. 1995, Proceedings National Academy of Sciences, U. S. A. 92: 9861-9865).

Line 174: Provide more information on the social and economic aspects of the producers.

Line 196-197: Provide more information on the social and economic constraints impeding the implementation of IPM practices.

Line 207: This section is missing information on the type of coffee growing in Brazil, which is important because they grow both C. arabica and C. robusta. Also, I suggest changing the title of this section because it is confusing, since it is not clear why in Brazil chemical control of CBB is not sustainable (and it implies that it is sustainable in other countries).

Line 211: Provide information on coffee cultivation and the social and economic characteristics of producers in Brazil.

Line 212: Reference 60 refers to India, but is cited for a text on Brazil. Better to use another reference, which is more appropriate for the case of Brazil.

Line 241-242: Provide more information on the social and economic constraints impeding the implementation of IPM practices, or explain why the losses were so high.

Line 251: Considering that the leaf miner is the main pest of coffee in Brazil, and it is controlled mainly with insecticides [e.g. Pantoja-Gomez et al. 2019. J. Econ. Entomol. 112 (2): 924-931], discuss possible effects on CBB in Brazil.

Line 256: Robusta and conilon are the same type of coffee.

Lines 266-267: Update the ICO reference (according to this reference, the information is from 2012, see line 1146).

Line 278: Is there an explanation for this difference in infestation between small-scale and large-scale plantations?

Line 323: Before this section 7, a paragraph or section is necessary to contrast the situation of the three cases presented above, highlighting the teachings or learnings in the management of CBB.

Line 325: Incorrect information. Coffee was not introduced to Puerto Rico from Spain (see https://issuu.com/coleccionpuertorriquena/docs/el_cafe_en_pr-saldana).

Line 343: Provide information on the types of producers as well as the ecological and management characteristics of the coffee farms.

Line 377: How many plots were sampled?

Line 405: The possible control of CBB by eliminating Wolbachia with antibiotics would be more of a type of chemical control, not biological control. If you have the opposite opinion, argue the case regarding the concept of biological control (see: Garcia, R., L.E. Caltagirone y A.P. Gutierrez. 1988. Comments on a redefinition of biological control. BioScience 38: 692-694.).

Line 412: Include “L.” (Linneo): Coffea arabica L. var. typica Cramer

Line 427-428: Include the names of the islands in Figure 2.

Line 510: Provide information on the constraints impeding the implementation of IPM practices.

Line 524: How can one explain that the colonization of Papua New Guinea by the CBB happened so long after, considering that the CBB was so close (and for so long) in Indonesia?

Line 557: Before this section 8, a paragraph or section is necessary to contrast the situation of the three cases presented above, highlighting the teachings or learnings in the management of CBB.

Line 565: Include a figure with the types of traps used in Hawaii.

Line 606: Describe the visual method used by the farmer

Line 630: Include labor costs.

Line 675: In general, how many sprays are done per year?

Line 789: What would be the challenges for insecticide research?

Line 947: The reference says 2015a, but there is no reference 2015b. Check it out.

Line 1079: The word “Divulgação” is misspelled.

Author Response

Line 18 and line 37: It says Ferrari, it must say (Ferrari).

We have made this correction.

Lines 40-41: In order to match the photograph (Fig. 1 B), it should be clarified that the insect bores the fruit (berry) of the coffee to later enter the seed (bean).

We have made this correction.

Line 40: Apparently CBB has not been reported in China either. Check it out.

Thank you, we have now included China as one of the few countries where CBB is not reported, yet.

Lines 74-75: It is not clear how these figures were obtained (in one of the countries mentioned there are more publications than those mentioned). Therefore, more detail is required on how the selection of the publications was made. It is desirable that the authors include the list of the mentioned publications as supplementary material.

We apologize for the lack of clarity in how Fig. 3 was obtained. We have changed and added some details to this paragraph to clarify how publications were selected. With thousands of publications on CBB we feel the inclusion of a list of all articles in the supplementary material is extraneous.

Line 75: In Fig. 3, indicate with an arrow (or other symbol) the year of the introduction of CBB to US coffee-producing regions.

Thank you for this suggestion, we have now included this in Fig. 3A.

Line 92: It is not clear which is the reference from which the data was obtained for 181 described species of the genus Hypothenemus.

We have added relevant references to support this statement.

Line 95: I didn’t find Waterhouse and Norris (1989) in the References section.

This citation has been added to the references.

Line 116: I didn’t find Hulcr et al. (2015) in the References section.

This citation has been added to the references.

Line 118: Cite the reference that the CBB was first reported in Liberia in 1897.

The relevant citation was added to support this statement.

Line 127: Probably the maximum altitude for robusta (C. canephora) in West and Central Africa of the mentioned range (250-1500 m), is incorrect. Check it out.

We have edited this to 250-1100 m.

Line 134: It is imprecise to say “when the dry matter content is 20% or greater”, for example, see Fig. 7, p. 74 in Barrera (1994) [Barrera, J.F. 1994. Dynamique des populations du scolyte des fruits du caféier, Hypothenemus hampei (Coleoptera: Scolytidae), et lutte biologique avec le parasitoide Cephalonomia stephanoderis (Hymenoptera: Bethylidae), au Chiapas, Mexique Ph.D. thesis. Université Paul Sabatier Toulouse, France, 301 pp] (http://www2.tap-ecosur.edu.mx/mip/TesisJFB/12_Chapitre_III_Floraison.pdf). This figure shows that before reaching 20% dry weight, when the coffee seed is not suitable for CBB to initiate oviposition, the dry weight is also greater than 20%, however, at this stage the endosperm of the seed is not yet has completed its development. According to this figure, it is more appropriate to say “when the dry weight content of the coffee seed is 20% and the endosperm is in the state of development known as semi-consistent”. This is true for the case of C. arabica.

We have revised this statement as suggested.

Line 136: In many cases, CBB also bored next to the floral disc, not only in its central part.

We have revised this statement as suggested.

Lines: 137-138: “…unless the infestation is high…” and the availability of berries is scarce.

We have revised this statement as suggested.

Line 165: Same as Line 40.

We have now included China as one of the few countries where CBB is absent.

Lines 147-148: I suggest mentioning the mode of sex determination of CBB, for example the “functional haplodiploidy” and discussing its importance in CBB colonization of new regions (Brun et al. 1995, Proceedings National Academy of Sciences, U. S. A. 92: 9861-9865).

We have added this information as suggested.

Line 174: Provide more information on the social and economic aspects of the producers.

This information has been added as suggested.

Line 196-197: Provide more information on the social and economic constraints impeding the implementation of IPM practices.

Thank you, we have made the statement clearer, and now include education as a factor (see comment from reviewer 3, below).

Line 207: This section is missing information on the type of coffee growing in Brazil, which is important because they grow both C. arabica and C. robusta. Also, I suggest changing the title of this section because it is confusing, since it is not clear why in Brazil chemical control of CBB is not sustainable (and it implies that it is sustainable in other countries).

Thank you, we have made the suggested edits.

Line 211: Provide information on coffee cultivation and the social and economic characteristics of producers in Brazil.

Thank you, we have added the suggested information.

Line 212: Reference 60 refers to India, but is cited for a text on Brazil. Better to use another reference, which is more appropriate for the case of Brazil.

We have made the change as suggested.

Line 241-242: Provide more information on the social and economic constraints impeding the implementation of IPM practices, or explain why the losses were so high.

We have now clarified this statement.

Line 251: Considering that the leaf miner is the main pest of coffee in Brazil, and it is controlled mainly with insecticides [e.g. Pantoja-Gomez et al. 2019. J. Econ. Entomol. 112 (2): 924-931], discuss possible effects on CBB in Brazil.

We have now included this information.

Line 256: Robusta and conilon are the same type of coffee.

Thank you for pointing this out, we have made the correction.

Lines 266-267: Update the ICO reference (according to this reference, the information is from 2012, see line 1146).

We have corrected this reference.

Line 278: Is there an explanation for this difference in infestation between small-scale and large-scale plantations?

We have now included this information.

Line 323: Before this section 7, a paragraph or section is necessary to contrast the situation of the three cases presented above, highlighting the teachings or learnings in the management of CBB.

We have now included a paragraph summarizing section 6 as suggested.

Line 325: Incorrect information. Coffee was not introduced to Puerto Rico from Spain (seehttps://issuu.com/coleccionpuertorriquena/docs/el_cafe_en_pr-saldana).

Thank you, we have corrected this statement.

Line 343: Provide information on the types of producers as well as the ecological and management characteristics of the coffee farms.

This information has been added as suggested.

Line 377: How many plots were sampled?

We have added this information as suggested.

Line 405: The possible control of CBB by eliminating Wolbachia with antibiotics would be more of a type of chemical control, not biological control. If you have the opposite opinion, argue the case regarding the concept of biological control (see: Garcia, R., L.E. Caltagirone y A.P. Gutierrez. 1988. Comments on a redefinition of biological control. BioScience 38: 692-694.).

Yes – the use of antibiotics to reduce or eliminate Wolbachia would be a sort of chemical control. However, manipulation of the bacteria or the CBB microbiome and its potential interference on CBB reproduction could be considered a biological control. We have changed this word to ‘control’ in this section for simplicity.

Line 412: Include “L.” (Linneo): Coffea arabica L. var. typica Cramer

We have added the species authority as suggested.

Line 427-428: Include the names of the islands in Figure 2.

We have edited Fig. 2 as suggested.

Line 510: Provide information on the constraints impeding the implementation of IPM practices.

This information is already included: poor farmer education, low labor availability, farmer resistance to performing sanitation picks and pruning due to already low yields, high production costs.

Line 524: How can one explain that the colonization of Papua New Guinea by the CBB happened so long after, considering that the CBB was so close (and for so long) in Indonesia?

To our knowledge there is not a satisfactory explanation of why colonization on PNG came relatively later. We have avoided speculating.

Line 557: Before this section 8, a paragraph or section is necessary to contrast the situation of the three cases presented above, highlighting the teachings or learnings in the management of CBB.

We have added a summary paragraph before section 8 as suggested.

Line 565: Include a figure with the types of traps used in Hawaii.

We have included a new figure (Fig. 5) that shows essential components of a successful IPM program for CBB, including monitoring flight activity using traps (Fig. 5A).

Line 606: Describe the visual method used by the farmer.

These are casual observations, with farmers looking for CBB entrance holes in berries, as opposed to actively using traps or the 30-tree sampling method to monitor infestation. This information has been included as suggested.

Line 630: Include labor costs.

Roughly $2/pound or $10/hr. This information is now included.

Line 675: In general, how many sprays are done per year?

It can vary from 0-12, but most growers surveyed spray between 3-8 times per season. This information is now included as suggested.

Line 789: What would be the challenges for insecticide research?

Thank you, we include research and other hurdles now.

Line 947: The reference says 2015a, but there is no reference 2015b. Check it out.

Thank you for pointing this out, we have corrected this error.

Line 1079: The word “Divulgação” is misspelled.

Thank you, this has been corrected.

Reviewer 2 Report

The submitted review by Johnson is well motivated, the structure is appropriate, and the manuscript is well written without missing any key details. The present paper compares and contrasts patterns of coffee berry borer (CBB) invasion and response across its extensive global distribution. Accidental introduction of CBB into new areas has resulted in establishment and spread of these notorious palm pests in distinctly non-native habitats. The work further compares infestation patterns, economic impacts, and management strategies across invaded ranges. Future management strategies could exploit new technological developments for CBB surveillance and new association biological control. The inclusion of policy makers and growers is mandatory.

It would be useful for the authors to address and compare this case to other similar invasions (e.g., palm weevils; https://doi.org/10.1007/s10340-018-1044-3). Adding this information could benefit the discussion.

I enjoyed reading this manuscript and I could not find any egregious errors. Overall a nice piece of work.

Author Response

It would be useful for the authors to address and compare this case to other similar invasions (e.g., palm weevils; https://doi.org/10.1007/s10340-018-1044-3). Adding this information could benefit the discussion.

Thank you for this valid suggestion, however, we feel this comparison falls outside the scope of the current review, which is already quite lengthy. Future reviews will incorporate comparisons across a range of crop pests.

Reviewer 3 Report

This article takes stock of old and new CBB invasions around the world and also looks at the particular case of Hawaii where research on IPM has made considerable progress over the last ten years. The originality of this manuscript is that it goes beyond the simple exploration of literature, the description of events and the listing of different control techniques.  It highlights many recent results and comments on their possible integration into integrated pest management. A few remarks have been made in the manuscript, but the only part that really needs to be reworked is the introduction. The bibliography section also needs to be revised in order to harmonise the presentation of the different references.

I encourage authors to resubmit the manuscript after revision.

Author Response

This article takes stock of old and new CBB invasions around the world and also looks at the particular case of Hawaii where research on IPM has made considerable progress over the last ten years. The originality of this manuscript is that it goes beyond the simple exploration of literature, the description of events and the listing of different control techniques.  It highlights many recent results and comments on their possible integration into integrated pest management. A few remarks have been made in the manuscript, but the only part that really needs to be reworked is the introduction. The bibliography section also needs to be revised in order to harmonise the presentation of the different references.

Thank you we have made the suggested changes to the introduction and fixed the presentation of references.

L36: Your introduction lacks coherence. In the first paragraph, you associate damage, biology and dispersion without any real link between these different aspects. In the second paragraph, you begin by linking "world coffee consumption" and "research on CBB", even though there is no direct link between the two. You continue with statistics of publications about the CBB which lead to a record for the USA. What is the relevance of these considerations for this article? In the third paragraph, you present the different themes of your article, but in the end you do not give it a clear objective.

Thank you, this is a helpful observation. We have added in some linking ideas and the introduction is now more coherent.

Figure 2: You could add the island of Martinique (French Overseas Department) where the CBB was first reported in 2012.

We have colored the island of Martinique in light green to show that CBB persists here. However, we do not include this island as a case study in the present paper, and thus do not highlight this region in red.

L81: Place the different items in the order in which they appear in sections 2, 3, 4 and 5.

Thank you for this suggestion, we have made this change.

L84: Wouldn't it be first detection and then invasion?

We respectfully disagree; a pest would first have to invade an area before it could be detected.

L167-170: This paragraph falls within the scope of the introduction as it presents elements of objectives for this article.

Agreed, we have incorporated this sentence into the introduction.

L177: You should provide specific details on Arabica and Robusta crops (growing areas, coffee production, levels of CBB infestation, etc.).

This information is now included.

L196: Not only that. You must add the lack of knowledge about the biology and dispersal of CBB, for most of coffee producers.

Thank you, we now mention educational constraints as well.

L199: It seems to me that this is not the right reference.

These are the correct references as stated.

L265: Skip a line

Done.

L266: A few words on the origin of coffee growing in Ethiopia.

Thank you, we have added this.

L285: Before presenting the issues involved in the fight against CBB in Africa, can you explain what is being done in Ethiopia?

We have clarified that the CBB management described for most African countries includes Ethiopia.

L313: What are these arguments?

We have made the sentence clearer to indicate that the arguments include higher CBB infestation and lower coffee yield.

L315: Is there any information on the quality of agronomic maintenance of the plots that could explain the differences observed?

This would be difficult to compare without a detailed meta-analysis, but we do now mention that agronomic differences between plots and studies could be significant.

L510: Change to "(56% in 2018) (A. Kawabata, pers. comm.)."

This change was made as suggested.

L584: It is not a percentage but a ratio.

Thank you for pointing this out, we have corrected this error.

L604: However, recent surveys....

Done.

L609: Can you cite bibliographical references?

Done.

L367: It's not very clear. I assume you mean CBB control during transport of the crop to the pulping facilities and during post-harvest activities. Explain that transport is done in bags.

We have clarified this sentence.

L763: Also consumes first instar larvae

We have now included this information.

L806: Beware of this conclusion. In this trial, the control (with sanitation harvesting) was not a true control because it was exposed to the surrounding infestations throughout the trial.

We respectfully disagree with the reviewer on this point; the control in this study was a true control. The aim of the study was to determine if exclusion netting could reduce CBB infestation in protected plots. The neighboring control plots were left unnetted as a comparison for baseline infestation levels with the netted treatments. Both the controls and netted plots received the same ground and tree sanitation prior to initiation of the study, and were equally vulnerable to infestation from CBB persisting in the surrounding tree/ground berries.

L857: Are you aware of CBB lineage that are more aggressive than others? What does the bibliography have to say about this? Isn't the term "lineage" more appropriate than "variety" which is more used for  plants?

We have edited this statement to improve clarity.

L869: Could you give precisions?

Yes, we have specified and no longer mention “similar measures”

L889: It is not only temperature that can influence infestation levels. Certain rainy episodes linked to climate change can cause abundant falls of berries during ripening and considerably increase CBB populations on the ground in previously infested berries.

Yes, although we agree that this likely occurs, there is also data that suggests heavy rains can reduce CBB populations on the ground due to fungal pathogens and flooding conditions that can increase CBB mortality.